EMBO
reports

# KAP1 regulates endogenous retroviruses in adult human cells and contributes to innate immune control

Christopher HC Tie[1], Liane Fernandes[1], Lucia Conde[2], Luisa Robbez-Masson[1], Rebecca P Sumner[1], Tom Peacock[1], Maria Teresa Rodriguez-Plata[1], Greta Mickute[1], Robert Gifford[3], Greg J Towers[1], Javier Herrero[2] (ID) & Helen M Rowe[1,*] (ID)

## Abstract

Endogenous retroviruses (ERVs) have accumulated in vertebrate genomes and contribute to the complexity of gene regulation. KAP1 represses ERVs during development by its recruitment to their repetitive sequences through KRAB zinc-finger proteins (KZNFs), but little is known about the regulation of ERVs in adult tissues. We observed that KAP1 repression of HERVK14C was conserved in differentiated human cells and performed KAP1 knockout to obtain an overview of KAP1 function. Our results show that KAP1 represses ERVs (including HERV-T and HERV-S) and ZNF genes, both of which overlap with KAP1 binding sites and H3K9me3 in multiple cell types. Furthermore, this pathway is functionally conserved in adult human peripheral blood mononuclear cells. Cytosine methylation that acts on KAP1 regulated loci is necessary to prevent an interferon response, and KAP1-depletion leads to activation of some interferon-stimulated genes. Finally, loss of KAP1 leads to a decrease in H3K9me3 enrichment at ERVs and ZNF genes and an RNA-sensing response mediated through MAVS signaling. These data indicate that the KAP1-KZNF pathway contributes to genome stability and innate immune control in adult human cells.

**Keywords** 5-azacytidine; adult human cells; endogenous retroviruses; epigenetic control; innate immune genes; KAP1 (KRAB-associated protein 1); retrotransposons; SETDB1
**Subject Categories** Chromatin, Epigenetics, Genomics & Functional Genomics; Immunology; Transcription

## Introduction

Retrotransposons occupy at least half of the human genome, and individual loci reflect the genetic conflict experienced by the host throughout evolution [1]. Retrotransposons are divided into two groups, those with long terminal repeats (LTR) and those without (non-LTR). LTR retrotransposons are known as endogenous retroviruses (ERVs) because they are derived from exogenous retroviruses. ERVs constitute around 9% of the human genome [2] with important roles in human health and disease. For example, LTRs of the primate-specific retrovirus, MER41, function as natural poised enhancers for a network of interferon-induced genes [3]. The co-option of ERVs into normal processes such as development and immune defense is the result of co-evolution of ERVs and their regulatory DNA sequences, which are scattered across the genome, with their hosts over millions of years [1].

Endogenous retroviruses are subject to transcriptional repression in adult tissues through DNMT1-dependent cytosine methylation [4]. This serves to prevent expression of their nucleic acids and proteins, which could potentially trigger an autoimmune response. Treatment of cancer cells with DNA demethylating agents (based on 5-azacytidine, 5-AZA) leads to the reactivation of ERVs [5–7] and induction of interferon-stimulated genes (ISGs). This signaling pathway proceeds through MDA-5, MAVS, and IRF7 and is thought to result from RNA sensing of double-stranded RNA derived from ERVs. This mechanism may contribute to anti-tumor immunity in patients treated with these drugs and remarkably, cancer initiating cells pretreated with 5-AZA form fewer tumors in mice in a MAVS-dependent manner [6]. Reactivation of ERVs in adult cells has also been linked to autoimmune diseases such as multiple sclerosis [8]. Despite the presence and importance of ERV transcriptional regulation in adult tissues, however, very little is known about the epigenetic control pathways in operation.

In contrast, it is well known that the KAP1 and krüppel-associated box domain zinc-finger protein (KZNF) pathway initiates epigenetic silencing at ERVs early in development [9–13]. KZNFs (known as KZFPs in mouse) recognize DNA target sequences mainly within repetitive DNA [14–16], and recruit KAP1 and epigenetic modifiers including HP1, SETDB1, and DNMTs [17–19]. KAP1

1   Division of Infection and Immunity, University College London, London, UK
2   Bill Lyons Informatics Centre, UCL Cancer Institute, London, UK
3   MRC-University of Glasgow Centre for Virus Research, Glasgow, UK
    *Corresponding author. Tel: +44 207 6796926; E-mail: h.rowe@ucl.ac.uk

has been shown to bind to certain ERVs in human CD4$^+$ T cells [11] and plays a functional role in ERV regulation in human development in neural progenitor cells [20]. KAP1 has been depleted in several differentiated murine cell types with little effect on ERVs [21,22]. SETDB1, in contrast, which exerts KAP1-dependent and KAP1-independent roles in ERV silencing [13], has an established role in repressing ERVs in somatic cells. It is augmented in certain cancer cells where it functions in immune evasion [23], and it is necessary for the silencing of ERVs in mouse B lymphocytes [24] and in mouse embryonic fibroblasts [25].

The potential role and relevance of KAP1 in differentiated cells, particularly in adult human tissues, is a fundamental and open question, which we set out to address here by employing CRISPR/Cas9-mediated genome editing.

# Results

## The human retrovirus HERVK14C is repressed by KAP1 in undifferentiated and differentiated cells including adult PBMCs

We selected the ERV lineage HERVK14C as a tool to explore KAP1 function in differentiated cells because it has been reported to be KAP1-regulated in undifferentiated human embryonic stem cells (ESCs) [11]. HERVK14C is restricted to Great Apes and Old World Monkeys (Fig 1A) and is a low copy number ERV (Fig EV1A), making it relatively easy to study. We designed two primer sets to detect specific HERVK14C loci (Fig 1B) and employed KAP1 depletion and qRT–PCR to confirm that KAP1 represses these ERVs in undifferentiated OCT4-expressing embryonic NTERA-2 cells (Figs 1C and EV1B). We then focused on two differentiated cell lines, HeLa and 293T cells, and generated KAP1 knockout clones using CRISPR/Cas9 genome editing. We validated KAP1 knockout and complementation functionally using described KAP1-KZNF reporters [9,26] (Fig EV1C and D), and we discovered that like in undifferentiated cells, KAP1 represses HERVK14C in differentiated cell lines (Fig 1D and E). Two other KAP1-regulated retrotransposons, L1PA4 (an L1 subfamily) and SVA D (a SVA subfamily), in contrast, were less affected by KAP1 depletion (Fig 1C–E). One possibility why SVAs may be harder to resurrect could be due to their enriched cytosine methylation (Fig EV1E). Most importantly, we found using shRNA-mediated RNAi that KAP1 is necessary to regulate ERVs not only in cell lines but also in adult peripheral blood mononuclear cells (PBMCs), and to a lesser extent in CD4$^+$ T cells (Figs 1F and EV1F–H), in which the HERVK14C 5′LTR was enriched for cytosine methylation (Fig EV1I). Overall, these data suggest that KAP1 regulation persists in adult human cells.

## A common role for KAP1 in repressing ERVs and ZNF genes

To gain insight into the role of KAP1 in differentiated cells, we selected KAP1 knockout HeLa cell clones (from Fig 1D) for mRNA-sequencing. Sequencing reads were mapped to the human genome and to RepBase to assess the global expression of genes and repetitive elements and we focused on significant changes (where differential effects were > 2-fold with *P*-values < 0.05). Results reveal that KAP1 knockout induces a very specific phenotype involving over-expression of ERVs and ZNF genes (Fig 2). ERVs (HERV-S, HERV-K, and HERV-T, all of which are derived from exogenous retroviruses [27,28]) were the only class of repetitive elements overexpressed in knockout HeLa cells (Fig 2A and Dataset EV1), and as expected, these ERV families were also bound by KAP1 according to ENCODE data (Fig 2A and Appendix Fig S1A). These ERVs are present in diverse primates and other mammals (Fig 2B). Of note, several retrotransposons were downregulated in knockout cells, which might represent indirect effects (Appendix Fig S1B). The fact that few ERVs were reactivated may relate to the low number of active ERV transcriptional units in the human compared with the mouse genome [29].

Interrogation of the transcriptome showed that KAP1 knockout also affects several hundred cellular genes (Fig EV2A and B, Dataset EV2). When we focused on upregulated genes (> 2-fold where $P_{adj}$ < 0.05), we found that strikingly, ZNF genes, of which there were 13 (9 of which were KZNFs) are the class of cellular genes most significantly overexpressed (Fig 2C, with the highest 6.81× overexpressed, Fig EV2C). All of these KZNFs are also direct KAP1-binding targets (using ENCODE data; Fig EV2C), illustrating that KAP1 plays a functional role in transcriptional regulation of these genes, a notion that was previously only hypothesized [30,31].

## Conserved KAP1 binding and H3K9me3 at ERVs and ZNF genes

Since KAP1 repression of ERVs and ZNF genes is a phenotype that has also been observed in ESCs [10,11], we hypothesized that KAP1

---

**Figure 1.  The human retrovirus HERVK14C is repressed by KAP1 in undifferentiated and differentiated cells including adult PBMCs.**

A    Schematic diagram showing the age of HERVK14C.

B    Venn diagram showing the chromosome copies detected by the two primer sets.

C    qRT–PCR expression of endogenous repeats following shRNA-mediated KAP1 depletion in NTERA-2 cells. Results were normalized to β2 microglobulin (*B2M*). KAP1 expression levels were verified by qRT–PCR and Western blot. A representative experiment of two experiments is shown. Two-tailed unpaired *t*-test *P*-values are as follows: 0.002300 (HERVK14C_2; "shControl" compared to "shKAP1" at time points Day 4 and Day 6 combined).

D, E    qRT–PCR expression of endogenous repeats following KAP1 knockout in HeLa (D) and 293T cells (E). Results were normalized to *B2M*. KAP1 expression levels were verified by qRT–PCR and Western blot. For (E), we found that clone I (also treated with KAP1 sgRNA) retained KAP1 protein expression (E, right panel), so we only used clones II, III, and IV to explore phenotype (E, left panel), which represented validated KAP1 knockouts. Two-tailed unpaired *t*-tests were done to compare the controls ("Hela WT" in panel D or "293T WT" in panel E) against all respective KO clones (4 HeLa KO clones in panel D or 3 293T KO clones in panel E). *P*-values are as follows: (D) HERVK14C_1: 0.0017, HERVK14C_2: 0.0044, SVA D VNTR: 0.0243; (E) HERVK14C_2: < 0.0001. Clones 8, 12, and 15 (from D) were selected for mRNA-sequencing. *n* = 4.

F    qRT–PCR expression of endogenous repeats following shRNA-mediated KAP1 depletion in PBMCs (day 6 post-transduction). Results were normalized to *B2M*. Unpaired *t*-tests: *P*-values: 0.0394 (L1PA4), 0.0079 (HERVK14C), 0.0009 (SVA D VNTR), 0.0033 (GAPDH), <0.001 (B2M). *n* = 3.

Data information: All error bars show standard deviation (SD). All numbers above bars depict fold changes compared to control cells (to one decimal place). ***P* < 0.001, ***P* < 0.01, and **P* < 0.05.

Source data are available online for this figure.

may bind ERVs and ZNF genes at common genomic loci in both undifferentiated and differentiated cells. To address this question, we determined whether any KAP1 binding sites were common between human ESCs and differentiated cells (293T cells) using public ChIP-seq data [11] and ENCODE data. We identified 614 common peaks (Fig EV2D and Dataset EV3). We found these loci to

be highly enriched for ERVs compared to their abundance in the genome (Fig 2D). We determined the nearest gene to each of these sites, and interestingly, gene ontology analysis showed that the most common gene cluster was ZNFs (of which there were 61, including 40 KZNFs; Fig 2E), mirroring our functional data for the upregulated genes (Fig 2C). Finally, examining the occurrence of LINE1

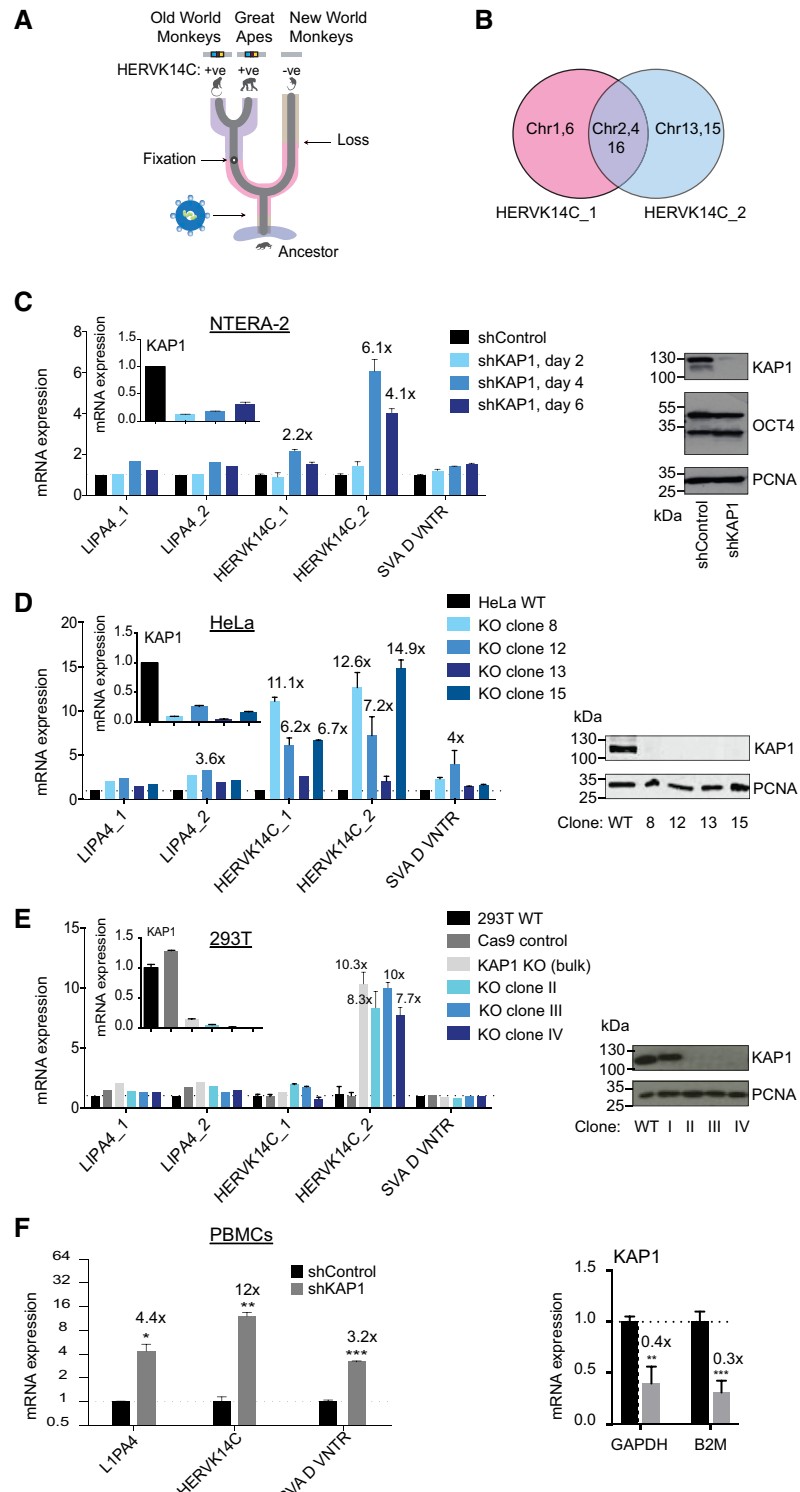

**Figure 1.**

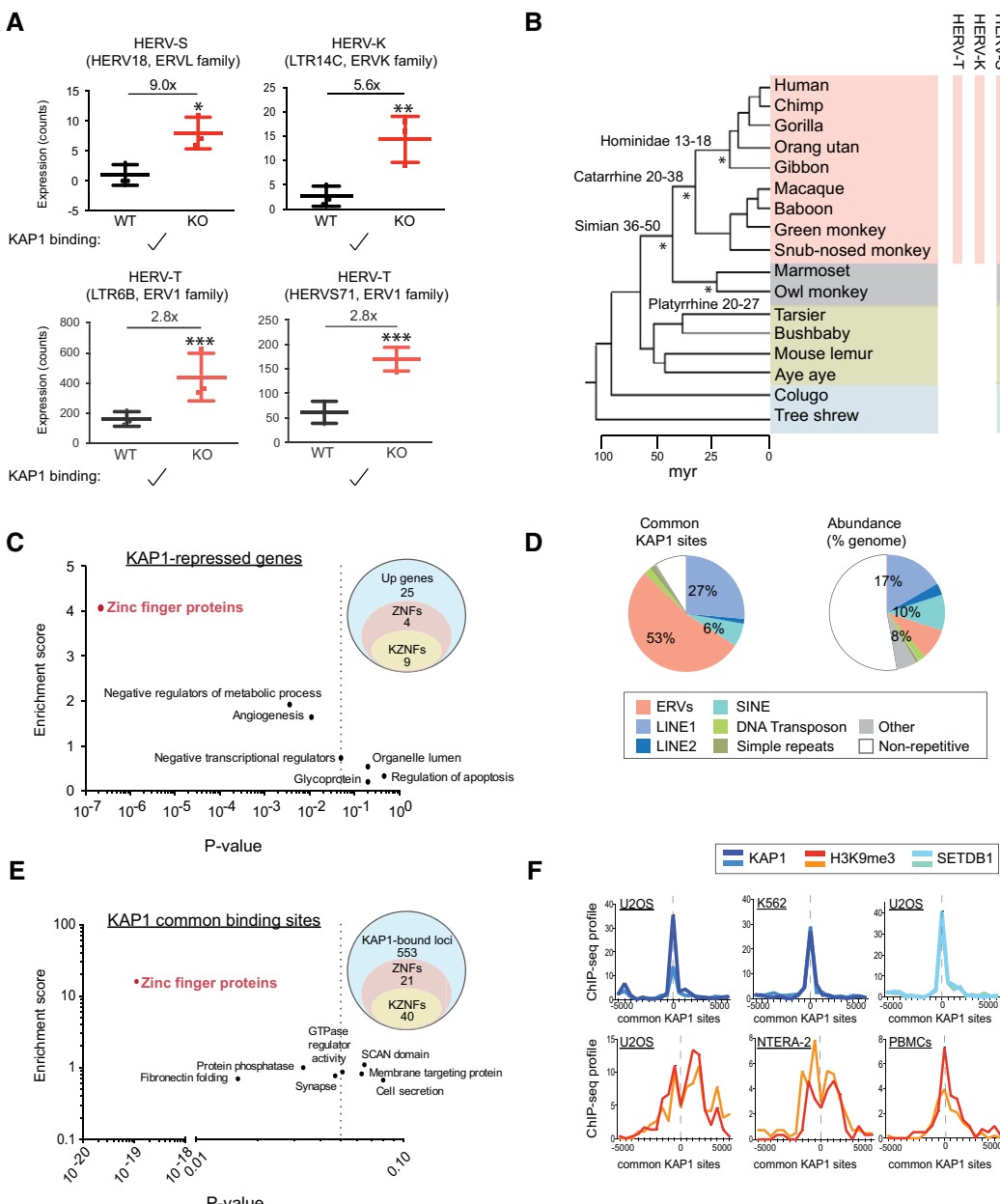

**Figure 2. A common role for KAP1 in repressing ERVs and ZNF genes through H3K9me3.**

A Boxplots showing repeats significantly upregulated (> 2-fold where *P* < 0.05 using DESeq2) in knockout (*N* = 3) compared to wild-type (*N* = 3) HeLa cells based on mRNA-sequencing data. *\*P* = 0.0174 (HERV-S), *\*\*P* = 0.0047 (HERV-K14C), *\*\*\*P* = 0.00013 (HERV-T LTR6B), and *\*\*\*P* = 1.90E-06 (HERV-T HERVS71). HERV-T and HERV-S but not HERVK14C also reached significance when only adjusted *P*-values were considered, where differences are compared to the whole of RepBase. KAP1 binding (ENCODE data) is shown in Appendix Fig S1A. All numbers above bars depict fold changes compared to control cells (to one decimal place). Error bars show SD.

B Evolutionary tree showing the age of KAP1-repressed HERVs identified in (A). Here, HERV-K refers to HERVK14C. Estimated ages of stated lineages are given and marked with a star. Myr: million years.

C The 38 upregulated genes identified (> 2-fold where $P_{adj}$ < 0.05) were converted to DAVID IDs and used for gene ontology analysis. Three gene clusters were significantly enriched (*P*-value < 0.05, drawn on the plot as a dotted line) in the data set: zinc-finger proteins ($P = 2.2 \times 10^{-7}$), negative regulators of metabolic processes ($P = 0.0036$), and angiogenesis ($P = 0.011$). Venn diagrams on the right show numbers of upregulated genes, ZNFs and KZNFs, and the overlap.

D The nature of conserved KAP1 binding sites between human ESCs [11] and 293Ts (ENCODE) (see Fig EV2D) is shown (left pie chart) compared to their abundance in the genome (right pie chart).

E The 614 KAP1 common binding sites (see Fig EV2D) were interrogated for their nearest gene, and this gene list was converted to DAVID IDs and used for gene ontology analysis. Four gene clusters were significantly enriched (*P*-value < 0.05, drawn on the plot as a dotted line): zinc-finger proteins ($P = 1.1 \times 10^{-19}$), fibronectin folding ($P = 0.016$), protein phosphatase ($P = 0.033$), and synapse ($P = 0.047$). Venn diagrams on the right show numbers of KAP1-bound loci, ZNFs and KZNFs, and the overlap.

F Genomic coordinates of the common KAP1 sites identified in Fig EV3D were subjected to ChIP-seq correlation analyses using ChIP-Cor software (see Materials and Methods). Each plot shows duplicate ChIP-seq experiments from ENCODE. See Fig EV3B for complete data.

elements and ERVs within KAP1 binding sites in (i) human ESCs or in (ii) 293T cells or (iii) within common sites revealed a similar pattern of distribution, except that ERV1 elements were slightly enriched within the common sites (Fig EV3A). This suggests that the landscape of KAP1 binding may be mainly conserved between these cell types. We then found using ENCODE data that common KAP1 sites correlate with the silent chromatin mark H3K9me3, as well as with KAP1 and SETDB1 in multiple cell types including primary adult cells (PBMCs; Figs 2F and EV3B for the whole dataset). There was no correlation with active histone marks (Fig EV3B).

### KZNFs expressed in differentiated human cells target ERVs

We reasoned that a subset of KZNFs must be widely expressed and at a sufficient level to recruit KAP1 to common DNA target sites. We therefore identified the top one hundred most highly expressed KZNFs in 293T cells using mRNA-sequencing data [32], verified them to be expressed at the protein level (the human protein atlas [33]), and recorded their targets where known [14,16] (Fig EV3C and Dataset EV4). Many of these KZNFs bind to ERVs (Fig EV3C), reflecting KAP1 binding at common loci (Fig EV3D). Of note, ZNF274 is widely expressed (Dataset EV4) and known to bind the 3′ ends of ZNF genes [30]. Interestingly, the ERV-binding KZNFs are widely conserved among primates or mammals and recognize ERVL or ERV1 sequences (Fig EV3E). These data show that KAP1 and KZNFs repress ERVs and ZNF genes in multiple cell types and suggest that core KZNF-ERV interactions may be conserved between species.

### PBS-dependent *de novo* repression of ERVs is conserved in primary adult cells

We next asked whether differentiated cells were equipped with high enough levels of KAP1 and its associated epigenetic machinery to induce *de novo* repression of incoming ERV sequences. KAP1/KZNFs are known to target several retroviruses in ESCs including HERV-K through the primer binding site (PBS) [11,12,34] as well as other sites [21,35]. We selected the KAP1-bound and repressed HERVK14C integrant on chromosome 15 as a model to measure *de novo* KAP1 repression in a reporter assay. This provirus contains a variant lysine PBS sequence (PBSChr15), which we cloned upstream of the HERVK14C LTR into a reporter vector (termed PBSChr15-LTR-GFP) and tested alongside a vector containing the consensus PBS-lys1,2 sequence (PBS-LTR-GFP) or a control vector with the HERVK14C LTR alone (LTR-GFP; Fig 3A). Using this system, we could demonstrate that the HERVK14C chromosome 15 PBS sequence and the consensus PBS sequence could induce KAP1-dependent reporter repression in both undifferentiated and differentiated cells (Fig 3B and C). Most importantly, we also detected PBS-dependent reporter repression in PBMCs, in which we verified that repression was not due to lack of reporter integration (Fig 3D). This suggested that a subset of KZNFs must be expressed in multiple differentiated cell types to recruit KAP1 to ERVs. To find out how many KZNFs are expressed at the mRNA level in HeLa cells, 293T cells, macrophages, and CD4[+] T cells, we analyzed mRNA-sequencing data and found 77 (Fig 3E, Dataset EV5) to be conserved. We verified that several KZNFs are also expressed at the protein level (Fig EV4).

### Cytosine methylation acts on KAP1-regulated retrotransposons and prevents innate immune activation

Endogenous retroviruses are subject to epigenetic repression in cancer cells, and disruption of cytosine methylation with anti-cancer 5-AZA-based drugs leads to immune activation [5,6], but it is unclear which epigenetic factors regulate 5-AZA-modulated ERVs. We asked whether KAP1, like cytosine methylation, is involved in controlling an intrinsic innate immune response by depleting KAP1 or treating cells with 5-AZA and measuring derepression of ERVs and induction of ISGs. We found that depletion of KAP1 alone leads to activation of the interferon-stimulated chemokines, CCL5 and CXCL10 in Hela cells but not in PBMCs (Fig 4A, Appendix Fig S2A and B). Of note, KAP1 depletion was less effective in PBMCs (Figs 1F and EV1F). In contrast, 5-AZA treatment leads to potent ISG induction in 293T cells, HeLa cells, and primary cells (Fig 4B, Appendix Fig S2C). 5-AZA is thought to act through reducing DNA methylation at retrotransposons rather than at ISGs. We verified that these ISG promoters were not methylated (ENCODE reduced representation bisulfite sequencing data and Ref. [6]) and found that 5-AZA treatment leads to SVA demethylation (Fig 4C). KAP1 depletion has little effect on SVA DNA methylation, perhaps explaining its only modest effect on ISGs (Fig 4C). The HERVK14C LTR, in contrast to SVAs, retains only low levels of DNA methylation that are little impacted by KAP1 depletion or 5-AZA (Appendix Fig S2D).

### KAP1 contributes to innate immune control

Once ISGs are activated, negative regulation of the interferon response prevents chronic immune activation [36]. Interestingly, we found that 12 immune response genes (Appendix Table S3) were downregulated in stable KAP1 knockout HeLa cell clones, suggesting that they may have been first activated. Of note, negative phenotypes would have been outselected during single-cell cloning of knockout clones and we found that initial KAP1 depletion caused a growth defect (Appendix Fig S2E) as described for SETDB1 depletion [25]. Therefore, we asked whether KAP1 depletion would lead to more potent immune induction in cells where reactivation of ERV nucleic acids and proteins is most pronounced. For this, we employed mRNA-sequencing data from naïve mouse ESCs depleted of epigenetic factors [37]. Indeed, we found that KAP1 depletion led to activation of genes involved in innate and adaptive immunity (Fig EV5A and B). SETDB1 depletion revealed similar results with both depletions overlapping in their activation of innate immune genes as the most significant gene cluster induced ($P = 0.0017$; Fig EV5C and D). Overall, these results suggest that KAP1 contributes to the regulation of retrotransposons and innate immune genes.

### KAP1 depletion leads to a decrease in H3K9me3 at ERVs and a MAVS-dependent innate immune response

A key question was how KAP1 is regulating ERVs and the innate immune response in differentiated human cells. We hypothesized that KAP1 regulates ERVs through H3K9me3 maintenance as is the case in ESCs [10,38]. We established that H3K9me3 was enriched at ZNF239, HERVK14C, and SVAs in both HeLa cells and 293T cells (Fig 5A). We then used 293T cells due to their amenity to genetic manipulation and depleted KAP1 or SETDB1 (Fig 5B, left), and

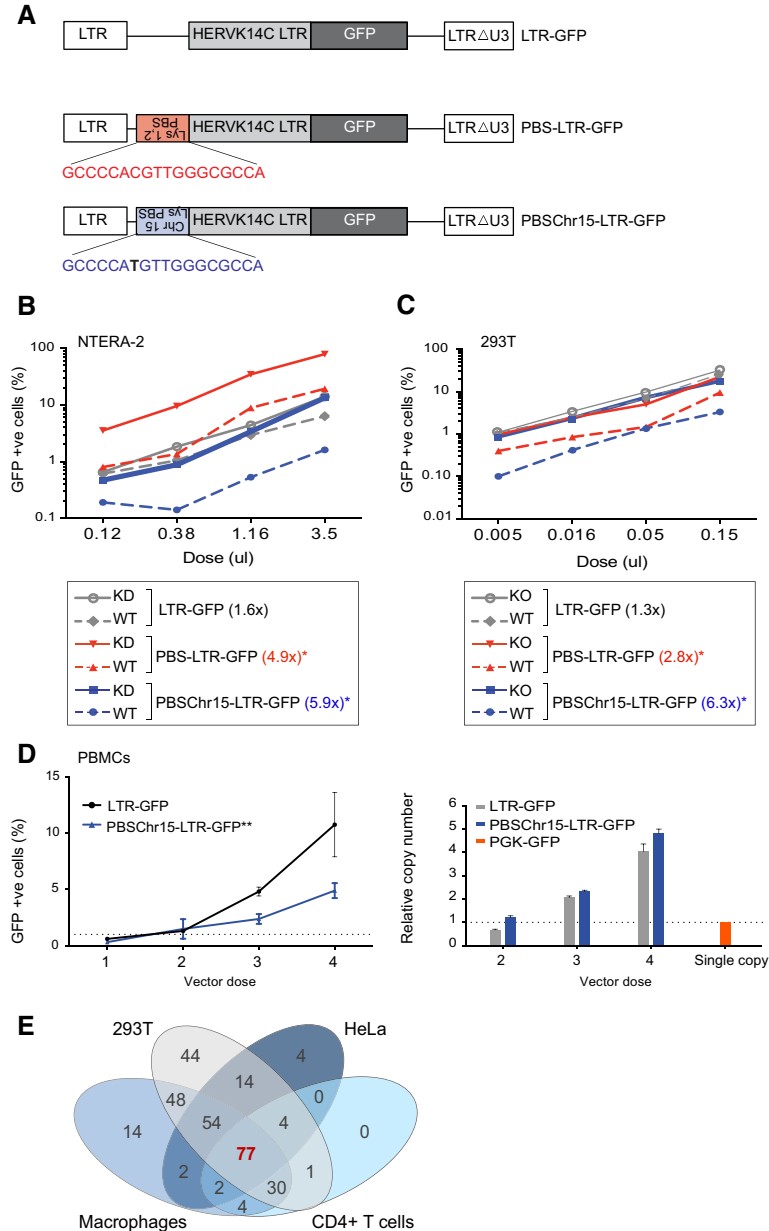

**Figure 3. PBS-dependent *de novo* repression of ERVs is conserved in primary cells.**

A  Schematic diagram showing reporter constructs used. The 18-bp PBS sequence was cloned upstream of the HERVK14C LTR promoter (which is identical in all vectors) in an antisense orientation.

B  NTERA-2 cells were transduced with either an empty vector (WT) or the same vector containing an shRNA against KAP1 (KD) prior to puromycin selection and transduction with increasing doses of GFP reporter vectors. GFP expression was analyzed 72 h post-reporter transduction. A representative experiment of two experiments is shown. Two-tailed unpaired *t*-test *P-values: PBS-LTR-GFP: 0.0148; PBSChr15-LTR-GFP: 0.0377.

C  The same as (B) except that here KAP1 wild-type (WT) and knockout (KO) 293T cells were used. *P-values: PBS-LTR-GFP: 0.0044; PBSChr15-LTR-GFP: 0.0077.

D  PBMCs were transduced with increasing doses of GFP reporter vectors (vectors were normalized to the same number of infectious units after titering them on permissive cells), left plot. Vectors were integrated at similar levels as measured by GFP Taqman qPCR, right plot. A 293T cell line with a single vector copy integrant (PGK-GFP) was used as a control to estimate the absolute copy numbers. A representative experiment of three experiments is shown here. Two-tailed unpaired *t*-test **P-value = 0.0042 (doses 3 and 4). Error bars show SD.

E  The expression of 340 KZNFs was assessed in multiple cell types using public mRNA-sequencing data, and Venn diagrams show the overlap in expression profiles. KZNFs were considered to be expressed in a particular cell type when their expression value was > 0.5 RPKM in all replicates.

found that either depletion led to a decrease in H3K9me3 at ZNF239 and HERVK14Cs, as well as HERVK14C reactivation (Fig 5B, right).

We next asked whether ISGs could be directly regulated by KAP1 and H3K9me3 by employing KAP1 and H3K9me3

ChIP-seq data [11,39]. For KAP1, we had identified 6,148 binding sites (Fig EV2D), and for H3K9me3, we found 18,271 enriched sites (common to duplicate ChIPs, Fig 5C, left). We identified 437 ISGs (genes induced 10-fold upon IFN treatment),

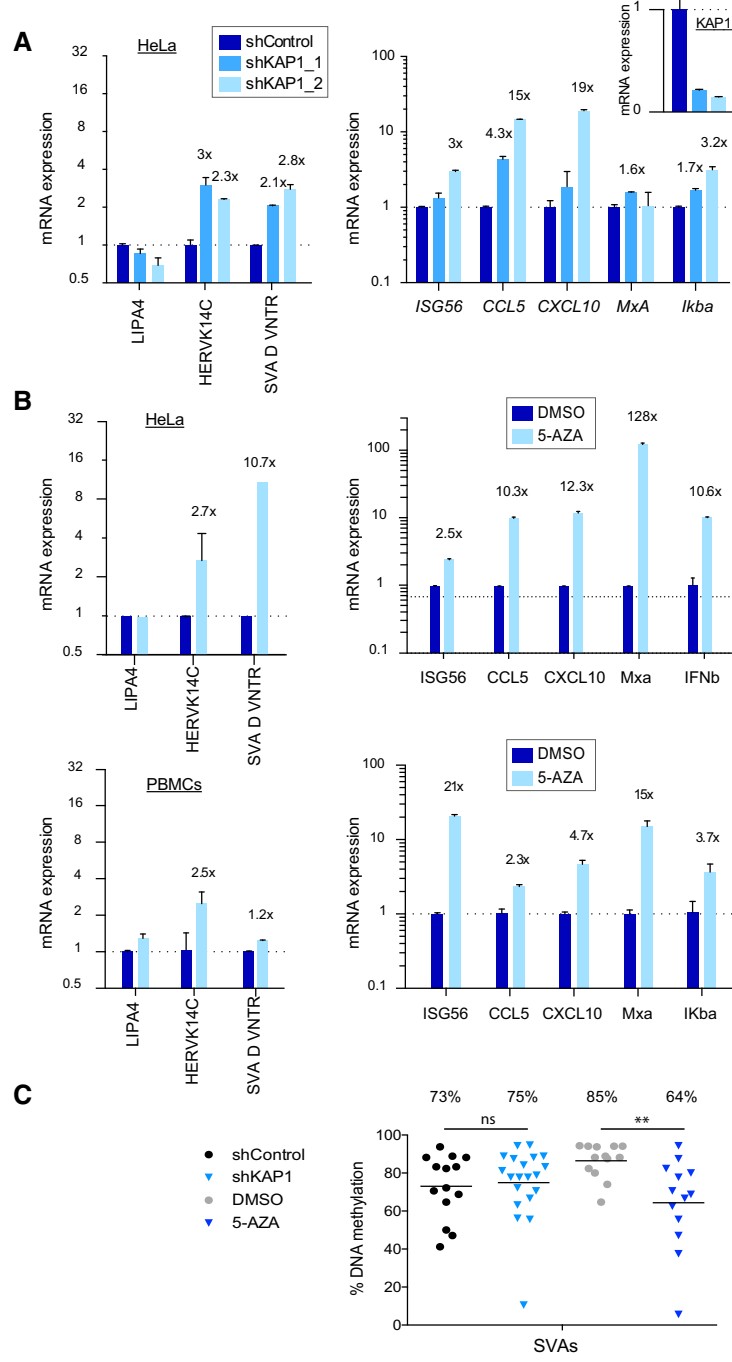

**Figure 4. Cytosine methylation acts on KAP1-regulated retrotransposons and prevents innate immune activation.**

A   qRT–PCR expression of endogenous repeats (left) and interferon stimulated genes (ISGs) (right) following shRNA-mediated KAP1 depletion in HeLa cells (day 6 post-transduction), normalized to *B2M* and *GAPDH,* respectively. *n* = 2. See also Appendix Fig S2A and B.

B   qRT–PCR expression of endogenous repeats (left) and ISGs (right) following 5-AZA treatment of HeLa cells and PBMCs (day 6 post-transduction). A representative experiment of two is shown in each case. Results were normalized to *GAPDH* and *B2M* (*GAPDH* results shown). *n* = 2. See also Appendix Fig S2C for results in 293T cells.

C   DNA methylation at endogenous SVAs was measured over 18 CpGs in the stated treatment groups (day 5 post-transductions, or day 2 post-5-AZA treatment). Each point represents the average methylation state of one sequence with at least 10 sequences analyzed per group. Mann–Whitney *U*-tests were performed to compare shControl to shKAP1 (*P* = 0.6593) and DMSO to 5-AZA (*P* = 0.0027). 5-AZA was added to all experiments at 7 μM.

Data information: All error bars show SD of technical triplicates. All numbers above bars depict fold changes compared to control cells (to one decimal place).

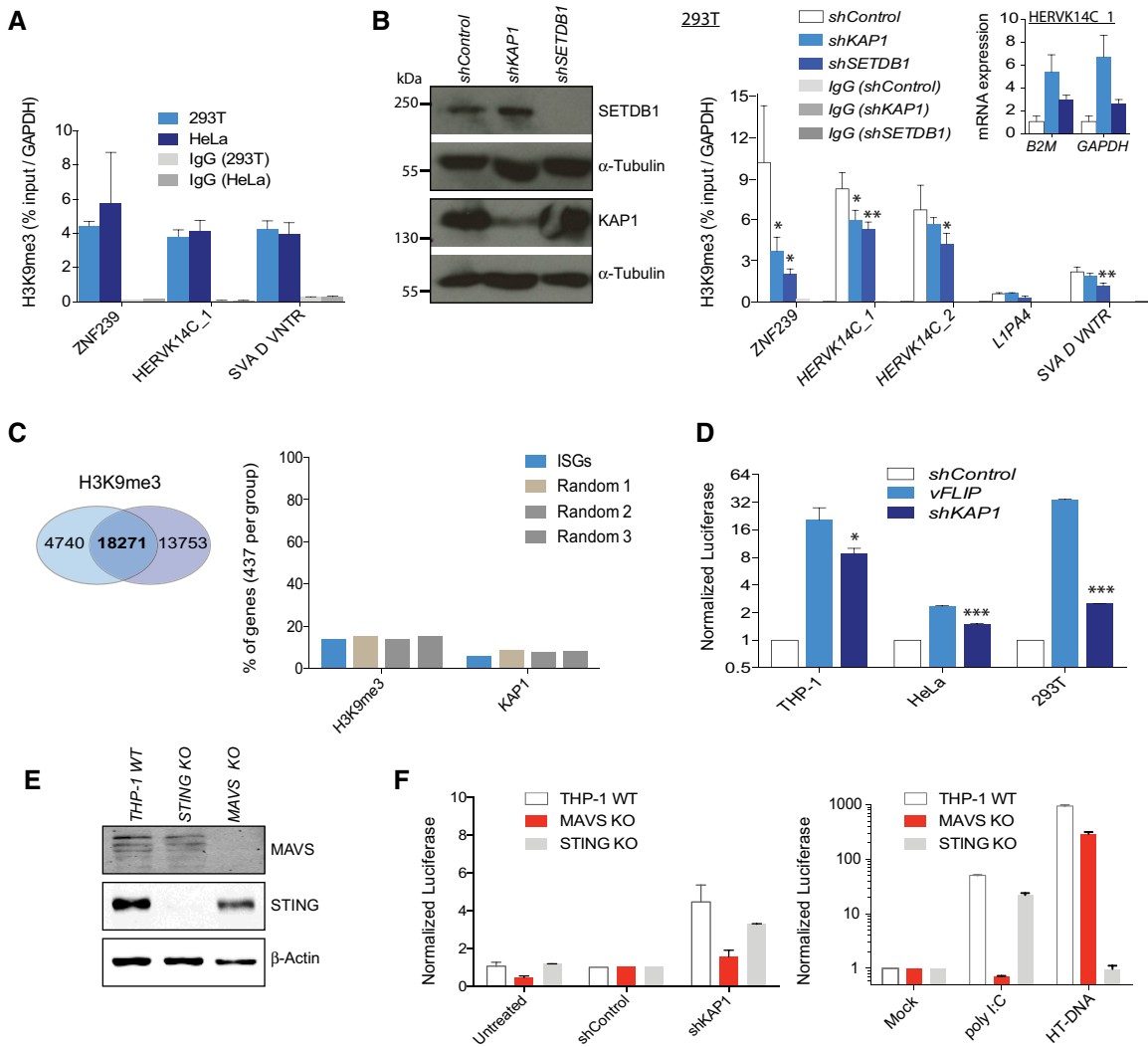

**Figure 5. KAP1 depletion leads to a decrease in H3K9me3 at ERVs and a MAVS-dependent innate immune response.**

A   H3K9me3 ChIP-PCR where error bars show SD of three independent IPs. Results show the IP enrichment relative to the total input normalized to *GAPDH* as a negative control region. IgG control IPs gave only background enrichments, shown.

B   293T cells were transduced with the stated shRNAs and cells harvested 4 days after puro selection for Western blot to verify KAP1 and SETDB1 depletion (left) and ChIP (right). Cell numbers were normalized per treatment group before sonication and IPs were performed in triplicate. Results show IP relative to total input and *GAPDH* normalized. IgG control IPs gave only background enrichments shown. One representative experiment of two is shown. Unpaired *t*-tests were performed. *P*-values: ZNF239: KAP1, 0.0292; SETDB1, 0.0137; HERVK14C_1: KAP1, 0.0227; SETDB1, 0.0082; HERVK14C_2: SETDB1, 0.0462; SVA D VNTR, SETDB1, 0.0064. The inlay shows qRT–PCR expression analysis of HERVK14C in the same experiment, normalized to *B2M* or *GAPDH* as stated. All error bars show SD.

C   Venn diagram showing H3K9me3-enriched sites (18,271) as those identified in both ChIP replicates in the human cell line K562 (left) using data from Ref. [39]. Right: % of ISGs or random genes intersecting a H3K9me3 or KAP1 ChIP-seq peak. 5 kb upstream and downstream of each gene was also included. *N* = 437 genes per group.

D   An *IFIT1* promoter-based luciferase reporter THP-1 cell line was used or 293T cells or HeLa cells transduced with an *IFNB* promoter-based luciferase reporter. These cell lines were transduced with an *shKAP1* or *shControl* vector or a *vFLIP*-expressing vector and luciferase measured 6 days later. Error bars show SD. Unpaired *t*-tests were used to compare *shControl* to *shKAP1* samples: *P*-values = 0.0122 (THP-1), 0.0001 (HeLa), 0.0001 (293T cells). *n* = 3.

E   WT, STING KO, and MAVS KO THP-1 cells were verified by Licor blot.

F   Cells from (E) were untreated or transduced with *shControl* or *shKAP1* vectors and luciferase read over time (left). Cells were also treated with control stimulants (right). Results show the average luciferase secreted over two time points (days 5 and 6 for the left plot and days 1 and 2 for the right plot) with all error bars showing SEM. HT-DNA: herring testis DNA. *n* = 2.

Data information: ***P* < 0.001, ***P* < 0.01, and **P* < 0.05.
Source data are available online for this figure.

using the tool: http://www.interferome.org [40] and also produced three independent sets of 437 randomly selected genes. These genes (including 5 kb each side) were intersected with the H3K9me3 or KAP1 peaks, which showed no significant differences of KAP1 (present on 6% of ISGs) or silent chromatin (H3K9me3, present on 14% of ISGs) per group (Fig 5C, right). This suggests that KAP1 does not regulate ISGs by directly binding to them. This is also important for understanding the mechanism of 5-AZA treatment of cancer since 5-AZA can impact on H3K9me3 deposition [41].

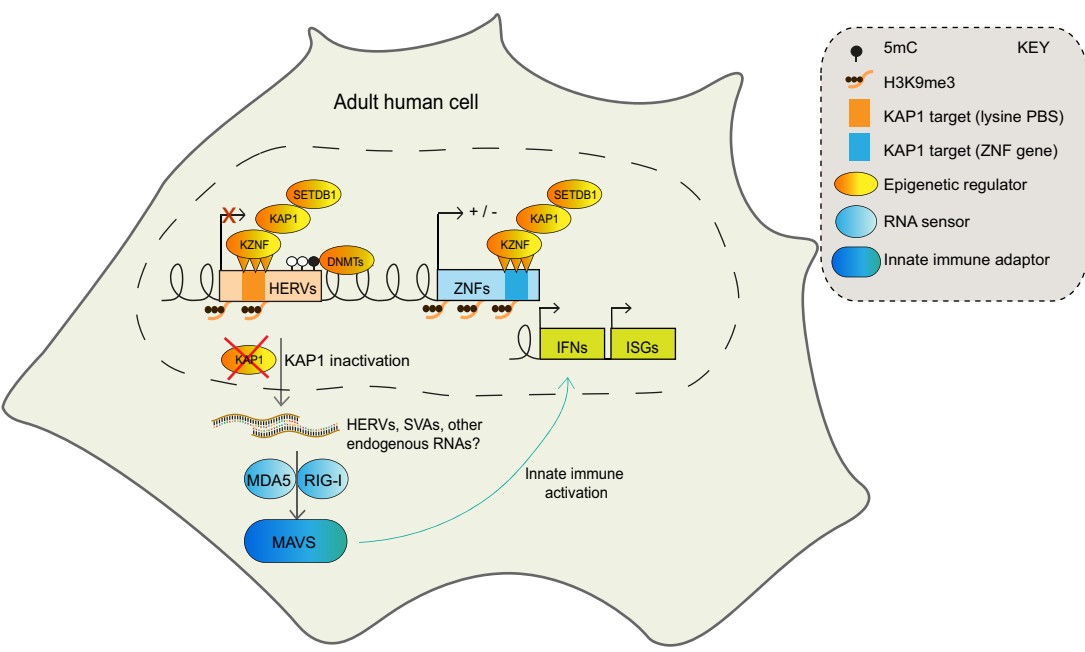

**Figure 6. Model.**

KAP1 regulates ERVs and ZNF genes in differentiated human cells including adult PBMCs and contributes to innate immune control. KAP1 and SETDB1 binding [30] are detected at ERVs and ZNF genes in differentiated cells and overlap with the silent chromatin mark H3K9me3 as well as some cytosine methylation, which we detect at HERVK14C and SVAs. KAP1 inactivation reveals that the KAP1-KZNF pathway is functionally intact and required in differentiated cells including PBMCs. KAP1 depletion leads to a decrease in H3K9me3 at retrotransposons and to reactivation of HERVs and SVAs. These retrotransposons or potentially other endogenous RNAs produce double-stranded RNA structures, which activate a type I interferon response through MAVS signaling. KAP1 depletion is not sufficient for global ISG induction, suggesting it exerts partial redundancy at repressing ERVs with other epigenetic mechanisms in differentiated human cells.

Finally, we asked whether the innate immune response activated upon KAP1 depletion is caused by an RNA-sensing or a DNA-sensing response. To address this question, we first selected THP-1 cells, in which we detected robust induction of a luciferase-based ISG reporter upon KAP1 depletion (Fig 5D). The reporter was *IFIT-1*-promoter-driven luciferase in THP-1 cells and *IFNβ*-promoter-driven luciferase in 293T and HeLa cells. Here, viral FLIP (vFLIP) (from KSHV) encoded from a lentiviral vector served as a positive control because it constitutively activates NFκB and the immune response [42]. We then employed CRISPR/Cas9 genome editing to disrupt the innate adaptors STING or MAVS in THP-1 cells (Fig 5E) and performed ISG reporter assays. MDA-5 and RIG-I RNA-sensing pathways are dependent on MAVS signaling [43], whereas the cGAS DNA-sensing pathway is dependent on STING [44]. Results revealed that ISG reporter induction observed upon KAP1 depletion is mainly dependent on MAVS signaling, consistent with an RNA-sensing innate immune response (Fig 5F). As expected, MAVS KO THP-1 cells were unable to mount a response to poly:IC but were competent for DNA sensing (of herring testis DNA, HT-DNA), whereas the reverse was true for STING KO THP1 cells (Fig 5F).

## Discussion

The present work highlights the previously unknown role for KAP1 in repressing ERVs and ZNF genes in differentiated human cells including adult cells, and in contributing to innate immune control.

KAP1 is relevant to the mode of action of anti-cancer 5-AZA-based drugs since KAP1 and cytosine methylation repress some of the same ERVs and suppress MAVS-dependent RNA sensing (see Fig 6 for a model). However, KAP1 knockout in differentiated cells impacts on ERVs and cellular genes only modestly compared to the more pronounced effects observed following KAP1 knockout in mouse or human ESCs [11,22]. This may relate to the increased requirement for KAP1 in early development to establish epigenetic marks [13]. Of note, the fact that 5-AZA exerts a dramatic effect on ISGs compared with KAP1 depletion could be due to a combination of effects this drug has on chromatin [41].

We found that a cohort of KZNFs are widely expressed at the mRNA and protein level in human cells and may restrict incoming retroviruses. Intriguingly, the mouse KZFP, ZFP809 is rapidly degraded at the protein level in mouse differentiated cells, restricting its potency mainly to ESCs [45]. An open question is whether KZNF expression is modulated by viruses, interferon treatment, or other stimuli as an antiviral strategy. Finally, it is possible that KAP1 repression of common loci is dynamic and KAP1-bound ERV regulatory sequences may have been co-opted to activate cellular gene networks. We found here that KAP1 regulates HERV-T and HERVK14C, both of which show evidence of co-option [27,46]. KAP1 is often enriched at regulatory hubs together with other transcription factors (Appendix Fig S1).

Future work will be necessary to determine exactly which factors act in addition to KAP1 to maintain ERV silencing in differentiated cells. Many epigenetic regulators exert cross-talk with

DNA methylation including SETDB1 [47], which can act independently or in concert with KAP1 [25] and which itself has a reported role in innate immune control in cancer cells [23]. Another key question concerns the identities of the potential immuno-stimulatory RNAs that may derive from retrotransposons producing secondary structure. Note, however, that to date there is no direct link known between ERV upregulation and an ISG response.

Hundreds of KZNFs have recently been described to target specific sequences within repetitive DNA, including ERVs [14–16]. The relevance of such interactions, however, remains unknown. Our work illustrates that cell lines rather than ESCs could be used to determine the individual functions of a core set of KZNFs that are widely expressed through their targeted gene knockout. Most interestingly, KAP1 and widely expressed KZNFs in combination with other epigenetic factors may become new drug targets for cancer with the aim to reactivate subsets of ERVs and other retrotransposons and harness their natural ability to trigger innate and adaptive anti-tumor immunity.

# Materials and Methods

### Cell culture

Human teratoma-derived NTERA-2 cells (kind gift from Peter Andrews, University of Sheffield) were cultured in Dulbecco's modified Eagle's medium (DMEM, Gibco) high glucose, supplemented with 2 mM L-Glutamine, 10% fetal calf serum (FCS), and 1% Penicillin/Streptomycin (P/S). They were split 1:2 or 1:3 every 3–4 days by cell scraping. HEK293T (293T) and HeLa cells were grown in standard DMEM + 10% FCS and P/S and split 1:4 every 2 days using trypsin. Human primary CD4$^+$ T cells were grown in Roswell Park Memorial Institute medium (RPMI, Gibco) supplemented with 10% human serum and 10 U/ml of recombinant IL-2 and activated using αCD3 and αCD28 pre-coated flasks for 72 h during which time IL-2 was increased to 25 U/ml. Peripheral blood mononuclear cells (PBMCs) were grown in RPMI supplemented with 20% FCS and 10 U/ml of IL-2 and activated using 3 μg/ml of Phytohemagglutinin-M (PHA) for 72 h. The THP-1 ISG reporter cell line (THP-1-IFIT-1-GLuc) was a gift from Veit Hornung. 5-azacytidine (5-AZA) was added at 7 μM or vehicle only (DMSO).

### CRISPR/Cas9 genome editing

Guide RNAs (sgRNAs, see Appendix Table S2) specific to several different exons of KAP1 (exons 1 and 9) were designed using the website: http://crispr.mit.edu/and cloned into the PX459 plasmid (Addgene), which was then transfected into HeLa and HEK293 T cells. After 24 h, the cells were subjected to puromycin selection for 24 h or until control cells had completely died. The bulk population was then used for single-cell cloning by limiting dilution. Knockout was assessed across a panel of clones by Western blotting using the KAP1 antibody, MAB3662 (Millipore), and validated functionally using KAP1-KZNF reporter assays. For knockout of STING or MAVS, sgRNAs (listed in Appendix Table S2) were cloned into lenti-CRISPRv2 and THP-1 cells transduced, selected with puromycin and single cell cloned as above.

### Primary cell isolation

peripheral blood mononuclear cells were isolated from a buffy coat or from fresh blood from healthy donors using lymphoprep (Axis-Shield). CD4$^+$ T cells were obtained from the isolated PBMCs using the CD4$^+$ T cell Isolation Kit (Miltenyi Biotec) according to manufacturer's instructions. The purity of the cells was verified by flow cytometry following antibody staining. Antibodies used are shown in Appendix Table S1.

### Western blotting

$1–2 \times 10^6$ cells were washed with PBS and lysed in NuPAGE LDS sample buffer (Thermo Fisher) with 5% β-mercaptoethanol. Samples were sonicated for 90 s at 20 Hz and heated at 95°C for 5 min. Lysates were then loaded onto handcast 10% SDS–polyacrylamide gels in tris/glycine/SDS buffer and mini-PROTEAN tanks (Bio-Rad), followed by wet transfers onto polyvinylidene difluoride (PVDF) membranes. Antibodies used for blotting the membrane are listed below. All secondary antibodies were horseradish peroxidase-conjugated (GE healthcare), and membranes were developed using ECL kits (ECL, Prime or Select kits from Amersham).

### shRNA lentiviral vectors and transduction

Hairpin sequences against human mRNAs were designed using the Clonetech RNAi designer website (http://bioinfo.clontech.com/rnaidesigner/) and annealed into oligo duplexes. The duplexes were then cloned into an shRNA vector (HIV SIREN) at *BamHI-EcoRI* sites, and the products were checked by sequencing. VSV-G-pseudotyped lentiviral vectors were produced by co-transfecting 293T cells in 10-cm plates with 1.5 μg of the shRNA plasmid, 1 μg p8.91, and 1 μg pMDG2 encoding VSV-G. The supernatant was harvested on two consecutive days (from 48 h post-transfection) and used neat or concentrated via ultracentrifugation ($20,000 \times g$ for 2 h at 4°C) for primary cells. For immune activation assays, vectors were pretreated with DNase to remove potential DNA from the vector preparations. Two days post-transduction, the cells were selected with puromycin (2.5 μg/ml) for 48 h (or until control cells had completely died) and qRT–PCR was done 6 days post-transduction.

### RNA extraction and quantification

Total RNA was extracted using an RNeasy mini kit (Qiagen) and DNase (Ambion AM1907) treated. 500 ng of RNA was used for cDNA production using SuperScript II Reverse Transcriptase (ThermoFisher) and random primers following the manufacturer's instructions. mRNA expression levels were quantified using quantitative reverse transcription PCR (qRT–PCR) using an ABI 7500 Real Time PCR System (Applied Biosystems). SYBR green Fast PCR mastermix (Life Technologies) was used. CT values for the test genes were normalized against those of *Gapdh* or *B2M* using the $-\Delta\Delta C_t$ method to calculate fold change. See Appendix Table S2 for primer sequences.

### DNA methylation analysis

DNA was harvested using a DNeasy Blood & Tissue Kit (Qiagen), and 1 μg of DNA was used for bisulfite conversion using an

EpiTect Bisulfite Kit (Qiagen) following the manufacturer's protocol. 4 µl of converted DNA was then amplified through PCR using the primer pairs described in the (Appendix Table S2). Primers were designed using the site: http://urogene.org/methprimer/and the PCR products were cloned using the TOPO TA-Cloning Kit (Thermo Fisher Scientific), and the T7P primer was used to sequence the products. DNA methylation status of the TOPO clones was measured using the QUMA online tool (http://quma.cdb.riken.jp) by the Riken Institute.

### GFP reporter assay

The HERVK14C LTR was cloned into a PGK-GFP plasmid in place of the PGK promoter at XhoI-BamHI sites. The consensus lysine PBS sequence was cloned upstream of the LTR after annealing primers into the XhoI site, while the Chromosome 15 HERVK14C-specific lysine PBS was cloned into the backbone through a PCR strategy into the XhoI-BamHI sites. The final products were verified via sequencing.

Cells were plated at a concentration of $10^5$ cells/ml in 24-well plates. After 6 h, wells were transduced with VSV-pseudotyped GFP vectors at increasing doses (normalized between vectors by the number of transducing units, which were calculated by first titering vectors on KAP1 knockout 293T cells) and fresh media was replenished after 24 h. After a further 48 h, the cells were fixed in 1% PFA and washed in PBS and GFP was read using flow cytometry.

### Intracellular OCT4 staining

$1 \times 10^6$ cells per condition were fixed and permeabilized using intracellular staining buffers (eBioscience, 00-5523). The cells were then stained with OCT4 or isotype control antibodies (see Antibody list above), washed, and analyzed by flow cytometry.

### Luciferase reporter assays

Luciferase assays were conducted according to the Promega Dual Luciferase Kit instructions. 293T cells were plated 6 h before transfection at a concentration of $10^5$ cells/ml in 24-well plates. Following ratios defined before [9], 200 ng of KRAB ZNF plasmid DNA, 20 ng of Luciferase reporter plasmid DNA, and 2 ng of pRT-TK_Renilla control plasmid were co-transfected (10:1:0.1 ratio) using 1.5 µl of Fugene 6 (Promega) and 30 µl of Opti-MEM (Gibco). Forty-eight hours post-transfection, cells were lysed and luciferase was measured using the Dual Luciferase® assay kit (Promega, E1910) in an opti-plate using a Glomax 96 microplate Luminometer's (Promega) Dual Glow program. The Renilla-encoding plasmid was used as an internal control for transfection efficiency normalization. The firefly to Renilla ratio was then further normalized against the empty vector or negative control as repression readouts and expressed in percentages where the control is set to 100%. The IFN-β promoter-driven F-luciferase lentiviral vector plasmid [48] was a kind gift from Jan Rehwinkel. For assays using the *IFIT1*-GLuc THP-1 cells, supernatant was harvested and coelenterazine (2 µg/ml, Prolume) was used as a substrate for luminescence readings. Controls were herring testis DNA (HT-DNA, 1 µg/ml, Sigma) and poly I:C (1 µg/ml Invivogen) and a vFLIP-expressing vector (kind

gift from Mary Collins). Data are presented as fold induction over the mock-treated sample for each cell line.

### mRNA-sequencing and analysis

Total RNA samples were processed using Illumina's TruSeq Stranded mRNA LT sample preparation kit (RS-122-2101) according to manufacturer's instructions with some deviations: Libraries to be multiplexed in the same run were pooled in equimolar quantities and calculated from Qubit and Bioanalyser fragment analysis. Samples were sequenced on a NextSeq 500 instrument (Illumina, San Diego, USA) using a 43 bp paired end run resulting in > 15 million reads per sample. Run data were demultiplexed and converted to fastq files using Illumina's bcl2fastq Conversion Software v2.16. Fastq files were then aligned to the human genome NCBI build 37.2 using Tophat 2.014 and then deduplicated using Picard Tools 1.79. Reads per transcript were counted using HTSeq, and differential expression was estimated using the BioConductor package DESeq2. *P*-values were adjusted for multiple testing with the Benjamini–Hochberg false discovery rate (FDR) procedure. Genes were considered upregulated or downregulated where differential affects were > 2-fold and where adjusted *P*-values were < 0.05. For analysis of repeats, TrimGalore v0.4.0 was used to remove adaptors and trim read ends using default parameters. Reads were mapped against the RepBase v20.06 human library using Bowtie2v2.2.4. The samtools v.1.19 idxstat utility was used to extract the number of mapped reads per repeat that were inputted to DESeq2 to identify differentially expressed repeats. *P*-values were adjusted for multiple testing with the Benjamini–Hochberg false discovery rate (FDR) procedure. Gene ontology analysis was conducted using the DAVID website: https://david-d.ncifcrf.gov/ [49,50]. Public mRNA-sequencing data were used to determine whether KZNFs were expressed in different cell types (see Data accessibility for accession numbers). A gene was considered to be expressed if it had an RPKM value of > 0.5 in all replicates. The top 100 KZNFs expressed in 293T cells were determined by sorting mRNA-sequencing data on RPKM values.

### Chromatin immunoprecipitation (ChIP)

ChIP experiments were performed as described [22] but using a Bioruptor Pico for sonication (four cycles, 15 s ON, 90 s OFF) and a H3K9me3 antibody (ab8898, Abcam). Chromatin immunoprecipitation-sequencing data for human ESCs and 293T cells were downloaded from NCBI Gene Expression Omnibus (GEO) under accession numbers GSE57989 (HuESC) and GSE27929 (293T). TrimGalore v0.4.0 was used to remove adaptors and trim read ends, and reads were mapped against the human genome (hg19 assembly) using Bowtie2 v2.2.4. Peaks were called in each replicate using Macs2 v 2.1.1, and the bioconductor package DiffBind (https://bioconductor.org/packages/DiffBind) was used to construct Venn diagrams and identify overlapping ESC and 293T peaks. Human repeat and gene locations were downloaded from the UCSC browser (RepeatMasker and RefGene tables), and the repeats and genes closest to the overlapping ESC/293T peaks were identified using bedtools v2.17.0. Chip-sequencing correlations were analyzed using the Chip-Cor website: http://ccg.vital-it.ch/chipseq/chip_cor.php.

## Statistical analysis

All error bars show standard deviation (SD) or standard error of the mean (SEM), as stated. Where $n \geq 3$, Student's *t*-tests or other statistical tests were used, as stated. A *P*-value of $< 0.05$ was considered statistically significant (\*\*\**P* $< 0.001$, \*\**P* $< 0.01$, and \**P* $< 0.05$).

## Ethics statement

Healthy adult blood donors provided written informed consent. Culture of primary peripheral blood mononuclear cells from blood donors has been reviewed and granted ethical permission by the National Research Ethics Service through The Joint UCL/UCLH Committees on the Ethics of Human Research (Committee Alpha) 2 December 2009; reference number 06/q0502/92.

## Data accessibility

mRNA-sequencing data are available on the NCBI Gene Expression Omnibus (GEO) (http://www.ncbi.nlm.nih.gov/geo/) database (accession number GSE114998), and other data are included in this article and its supplementary information files. Accession numbers for the public data are as follows: 293T: GSE27929 (KAP1 ChIP-seq), HuESC: GSE57989 (KAP1 ChIP-seq, mRNA-seq), 293T: GSE44267 (mRNA-seq), macrophages: GSE36952 (mRNA-seq), CD4$^+$ T cells: GSE69549 (mRNA-seq), K562 cells: GSE95374 (H3K9me3 ChIP-seq), Naïve mESCs: GSE107840 (mRNA-seq).

**Expanded View** for this article is available online.

## Acknowledgements

We thank Connor Husovsky for technical assistance and Steen Ooi, Pierre Maillard, and members of UCL Cruciform wing 3.3 for advice. We thank Peter Andrews for the NTERA-2 cells; Didier Trono for reagents previously generated in his laboratory; David Haussler for the SVA, LINE1, ZNF91, and ZNF93 constructs; and Mary Collins for the vFLIP-expressing vector. We also thank the UCL Genomics Unit and Tony Brooks for carrying out the mRNA-sequencing library preparation, sequencing, and initial data analysis. This work was supported through a UCL Grand Challenges Studentship awarded to CHCT (code 518099) and a Sir Henry Dale Fellowship, which is jointly funded by the Wellcome Trust and Royal Society (Grant number 101200/Z/13/Z) awarded to HMR. RG is funded by the MRC (MC_UU_12014/10) and GJT, RS, TP, and MTR by Wellcome Trust (108183) and ERC (339223) grants awarded to GJT. LF is funded through an ERC grant (678350) awarded to HMR. Lucia Conde and Javier Herrero were funded through a Cancer Research UK-University College London (CRUK-UCL) Centre Award [C416/A25145]. The authors acknowledge the use of the UCL Legion High Performance Computing Facility (Legion@UCL), and associated support services, in the completion of this work.

## Author contributions

CHCT conceived, designed and performed experiments, analyzed the data, and wrote the paper. LF, LR-M, RPS, TP, MTR-P, GM, RG, and GJT contributed to reagents, experiments, and ideas. LC and JH performed and conceived bioinformatics analyses. HMR conceived the study, performed experiments, analyzed data, and wrote the paper. All authors read and approved the final manuscript.

## Conflict of interest

The authors declare that they have no conflict of interest.

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
