## [Review Process File · EMBO Reports]

KAP1 regulates ERVs in adult human cells and contributes to innate immune control

Christopher HC Tie, Liane Fernandes, Lucia Conde, Luisa Robbez-Masson, Rebecca P Sumner, Tom Peacock, Maria Teresa Rodriguez-Plata, Greta Mickute, Robert Gifford, Greg J Towers, Javier Herrero, Helen M Rowe

Review timeline:	Submission date:	11 August 2017
	Editorial Decision:	17 August 2017
	Revision received:	17 May 2018
	Editorial Decision:	11 June 2018
	Revision received:	21 June 2018
	Accepted:	9 July 2018

Editor: Achim Breiling

Transaction Report: This manuscript was transferred to *EMBO reports* following review at *The EMBO Journal*

1st Editorial Decision

17 August 2017

Thank you for the transfer of your research manuscript to EMBO reports. I now went through the referee reports and the comments of the external advisor from The EMBO Journal (which I copied again below).

I think your study would be suitable for EMBO reports, but only after substantial revision. We would require that the referee concerns and the points of the advisor from The EMBO Journal are addressed experimentally, in particular to answer the question how KAP1 depletion does promote innate immune activation. The first part of the manuscript (up to the data in Fig. 5) needs to be shortened, as the referees and the advisor agree that the data is not novel. Thus, we would need a shortened and enhanced manuscript on the novel functional role of KAP1 in immune activation via ERVs.

Given the constructive referee comments, we would like to invite you to revise your manuscript with the understanding that all concerns of the referee and the advisor must be fully addressed in the revised manuscript (as detailed above) and in a complete point-by-point response. Acceptance of your manuscript will depend on a positive outcome of a second round of review. It is EMBO reports policy to allow a single round of revision only and acceptance or rejection of the manuscript will therefore depend on the completeness of your responses included in the next, final version of the manuscript.

Revised manuscripts should be submitted within three months of a request for revision; they will otherwise be treated as new submissions. Please contact us if a 3-months time frame is not sufficient for the revisions so that we can discuss the revisions further.

Please refer to our guidelines for preparing your revised manuscript:

<http://embor.embopress.org/authorguide#manuscriptpreparation>

Supplementary/additional data: The Expanded View format, which will be displayed in the main HTML of the paper in a collapsible format, has replaced the Supplementary information. You can submit up to 5 images as Expanded View. Please follow the nomenclature Figure EV1, Figure EV2 etc. The figure legend for these should be included in the main manuscript document file in a section called Expanded View Figure Legends after the main Figure Legends section. Additional Supplementary material should be supplied as a single pdf labeled Appendix. The Appendix includes a table of content on the first page, all figures and their legends. Please follow the nomenclature Appendix Figure Sx throughout the text and also label the figures according to this nomenclature. For more details please refer to our guide to authors.

Important: All materials and methods should be included in the main manuscript file.

Regarding data quantification and statistics, can you please specify the number "n" for how many experiments were performed, the bars and error bars (e.g. SEM, SD) and the test used to calculate p-values in the respective figure legends? This information must be provided in the figure legends. Please provide statistical testing where applicable.

We now strongly encourage the publication of original source data with the aim of making primary data more accessible and transparent to the reader. The source data will be published in a separate source data file online along with the accepted manuscript and will be linked to the relevant figure. If you would like to use this opportunity, please submit the source data (for example scans of entire gels or blots, data points of graphs in an excel sheet, additional images, etc.) of your key experiments together with the revised manuscript. Please include size markers for scans of entire gels, label the scans with figure and panel number, and send one PDF file per figure or per figure panel.

- a complete author checklist, which you can download from our author guidelines (<http://embor.embopress.org/authorguide#revision>). Please insert page numbers in the checklist to indicate where the requested information can be found.
- a letter detailing your responses to the referee comments in Word format (.doc)
- a Microsoft Word file (.doc) of the revised manuscript text
- editable TIFF or EPS-formatted single figure files in high resolution (for main figures and EV figures)

In addition I would need from you:

- a short, two-sentence summary of the manuscript
- two to three bullet points highlighting the key findings of your study
- a schematic summary figure (in jpeg or tiff format with the exact width of 550 pixels and a height of about 400 pixels) that can be used as part of a visual synopsis on our website.

I look forward to seeing a revised version of your manuscript when it is ready. Please let me know if you have questions or comments regarding the revision.

REFeree REPORTS

Referee #1:

Regulation of ERVs is important for ensuring transcriptional and genomic integrity of cells. Many ERV classes are kept at a very low transcription level by synergistic activity of silencing pathways that establish H3K9me3 and DNA methylation across ERVs. Kap1 is an important component of the silencing machinery linking KRAB-ZNF Proteins (which bind ERVs through their ZnF domains) with the Setdb1 histone methyltransferase to establish H3K9me3.

In their manuscript the authors investigate the function of Kap1 in ERV repression in different

human cancer cells lines and peripheral blood mononuclear cells (PBMC). They identify the ERV family HERVK14C and some Zinc Finger Proteins as Kap1 targets. They further show that these targets display presence of repressive and lack of active modifications in different cell types. Then the authors identify the PBS of HERVK14C as KAP1-dependent repression initiation site which can induce silencing in different cell types. Finally, they describe that DNA methylation is important for HERVK14C silencing in different cell types. ERV derepression upon treatment with DNA methylation inhibitors coincides with activation of interferone regulated genes, suggesting that active ERVs trigger innate immune activation.

This study confirms the importance of Kap1 in ERV silencing but fails to provide novel mechanistic insight. It is already known that Kap1 can target ERVs through KZFPs, but which ZNF would be specific for HERVK14C is unclear. It is also known that Kap1 binding coincides with repressive modifications, such as H3K9me3 and DNA methylation, but further insight into how these processes are linked is not provided. It is also known that derepression of ERVs can coincide with activation of the innate immune response, but further details of how this pathway is triggered are not provided. Therefore, due to the lack of novelty and mechanistic insight my feeling is that this manuscript is not really suited for EMBO Journal.

Referee #2:

In this manuscript, the authors described that the human endogenous retrovirus (ERV) HERVK14C is repressed by KAP1 in undifferentiated and differentiated cell lines. Furthermore, this pathway is functionally conserved in primary human peripheral blood mononuclear cells (PBMCs). They finally showed that cytosine methylation plays a role for silencing of KAP1-regulated ERVs and preventing production of immunostimulatory nucleic acids of the derepressed ERVs. Collectively, the authors proposed that the KAP1-KZNF pathway has evolved to play an important functional role in genome integrity and the control of viral mimicry in differentiated human cells. This is the first report to show that HERVK14C is derepressed by KAP1 knock down (KD) or knock out (KO) in differentiated human cells. Also, this is a novel finding that KD of KAP1 induces activation of interferon-stimulated genes (ISGs) in some human differentiated tumor cell lines. However, some of their evidences are not enough to support their entire ideas, such as how DNA methylation is crucial for KAP1-mediated HERVK14C silencing in PBMC or other tumor cell lines used in this study. Therefore, the reviewer requests to tackle following points to improve this manuscript for publication.

Major criticisms,

1. DNA methylation-KAP1 silencing issue

Bisulfite sequencing analysis should be done on the HERVK14C KAP1 binding or LTR regions in KAP1KO HeLa or 293T cells (Fig 1 E and F samples), and KAP1 KDed PBMCs and CD4+ T cells (Fig 5A and Fig S4C sample). DNA methylation and H3K9me3 analysis also should be done on the reporter constructs shown in Fig. 4B, C and 5B. Bioinformatics analysis showed that KAP1-targeted sites are enriched for H3K9me3 and SETDB1 (Fig 3A), but it is not demonstrated that how these epigenetic layers are functionally organized by KAP1 in differentiated cells. Once such data is available, the authors can address more mechanistic issue how KAP1 epigenetically represses ERV in differentiated cells.

2. ISG activation issue

KD of KAP1 induced ISG activation in HeLa cells but not in PBMCs. However, KAP1-targeted ERVs such as HERVK14C and SVA D VNTR were derepressed in both KAP1 KDed and 5-aza treated PBMCs (Fig 5A and Fig6B bottom panels), derepression level was even less in 5-aza treated ones. Thus, this data is inconsistent with the authors' simple story "5-aza treatment induces derepression of the KAP1-targeted ERVs by removing DNA methylation on those KAP1-targeted ERVs, then transcript of the derepressed ERVs induces ISG activation through ERV-derived nucleic acid sensing" shown in Fig 7. Thus, 5-aza has additional effect other than ERV derepression for the ISG activation. The authors should describe their results more carefully.

Minor points,

3. Fig. S2B,

- The authors should discuss the mechanism of how these elements are repressed by KAP1 KO.
4. RNA expression analysis of shKAP1 treated cells, the authors should describe when samples are harvested after KD.
 5. Fig. S4A, right panel. Is this KAP1 KD efficiency data? If so, no significant reduction in this case based on B2M as a standard? The authors should clarify.
 6. Fig. 5D This data should be shown in supplemental fig since just protein expression data does not mean for function of them in KAP1-mediated ERV silencing.
 7. Fig. 6A left panel. Font size of the ERV labels is inconsistent with other panels.

External expert advisor:

The general notion of KAP1 as a transcriptional repressor of ERVs is well-established and in fact recent papers have also revealed the role of this pathway in somatic cell types. The current work fills up a few details but up through Figure 5, we do not gain much new insight. The binding of KAP1 to the ZNFs (more specifically to their gene bodies, see below) is not too surprising since H3K9me3 has been reported in the gene bodies of these genes previously; in fact the specific ZFPs that bind these gene bodies and direct H3K9me3 have already been identified.

The implications of the data shown in Figure 6 on the other hand are quite exciting in my view. The fact that depletion of KAP1, like inhibition of DNA methylation, also promotes activation of INF-stimulated chemokines, at least in HeLa cells, indicates that activation of LTR elements via the KAP1/Setdb1 pathway may also promote "viral mimicry"/immune activation, including of some of the same ERVs induced with 5-AZA treatment. What is not addressed here is the role of dsRNAs, MDA-5, MAVS or IRF7 in this pathway/the activation of ISGs. So, I think this area could be developed further to add to the novelty of the study. Put simply, how does KAP1 depletion promote innate immune activation?

This paper from Farnham and colleagues - "5-azacytidine treatment reorganizes genomic histone modification patterns", should be cited, as it reveals that 5-AZA has effects on expression beyond direct regulation of DNA methylation, including disrupting H3K9me3 deposition...

Regarding the KZNF gene upregulation, it would be nice if the authors explicitly stated the levels of de-repression of these genes. There is conflict in the literature about the role of KZNF/KAP1 in repression of KZNF genes, as the levels of de-repression have generally been reported to be low or undetectable (ie no change), and the binding sites are generally in the gene bodies of the KZNF genes (actually interestingly enough over the regions coding the zinc-fingers themselves), rather than in the promoter regions of these genes, which the authors do not comment on. For the KZNF genes that appear to be regulated by KAP1, does KAP1 binding, or H3K9me3, spread to the promoter regions of these genes?

1st Revision - authors' response

17 May 2018

Referee #1:

Regulation of ERVs is important for ensuring transcriptional and genomic integrity of cells. Many ERV classes are kept at a very low transcription level by synergistic activity of silencing pathways that establish H3K9me3 and DNA methylation across ERVs. Kap1 is an important component of the silencing machinery linking KRAB-ZNF Proteins (which bind ERVs through their ZnF domains)

with the Setdb1 histone methyltransferase to establish H3K9me3.

In their manuscript the authors investigate the function of Kap1 in ERV repression in different human cancer cells lines and peripheral blood mononuclear cells (PBMC). They identify the ERV family HERVK14C and some Zinc Finger Proteins as Kap1 targets. They further show that these targets display presence of repressive and lack of active modifications in different cell types. Then the authors identify the PBS of HERVK14C as KAP1-dependent repression initiation site which can induce silencing in different cell types. Finally, they describe that DNA methylation is important for

HERVK14C silencing in different cell types. ERV derepression upon treatment with DNA methylation inhibitors coincides with activation of interferon regulated genes, suggesting that active ERVs trigger innate immune activation.

Q1: This study confirms the importance of Kap1 in ERV silencing but fails to provide novel mechanistic insight.

>Response: The main aim of this study was to provide evidence that the KAP1/KZNF pathway is functional and biologically relevant in human adult tissues, rather than dissecting mechanism. Indeed, this present work is the first to address the role of KAP1 in differentiated human cells (see also the response to Q1 of the advisor above). Nevertheless, we have also now addressed the mechanism of KAP1-regulation of ERVs in differentiated cells and of KAP1-regulation of the innate immune response as detailed below.

Q2: It is already known that Kap1 can target ERVs through KZFPs, but which ZNF would be specific for HERVK14C is unclear.

>Response: Our aim in this work was not to discover the KZNF(s) that can target HERVK14C. We detect HERV-S and HERV-T to be KAP1 regulated in human cells as well as HERVK14C so we expect that a whole group of KZNFs are going to be important to target KAP1 to different retroviral sequences. Indeed, we detected 77 KZNFs to be commonly expressed across a range of human cell types at least at the mRNA level (Figure 3E). We agree with the reviewer that uncovering which of these KZNFs are important in differentiated cells is a key question, which we attend to address through a screen in a follow-up study.

Q3: It is also known that Kap1 binding coincides with repressive modifications, such as H3K9me3 and DNA methylation, but further insight into how these processes are linked is not provided.

>Response: This study is not aimed at addressing new insight into processes linking H3K9me3 and DNA methylation. However, here we have now addressed the potential role of KAP1, which is our focus, in maintaining H3K9me3 and / or DNA methylation at retrotransposons in differentiated cells. We find that KAP1-depletion has little impact on DNA methylation in contrast to 5-AZA treatment that induces demethylation of SVAs, as expected (see Figure 4C). We do, however, identify a role for KAP1 in maintenance of H3K9me3 at retrotransposons (see Figure 5BC), consistent with what has been shown in embryonic cells (*Rowe et al., Nature 2010*).

Q4: It is also known that derepression of ERVs can coincide with activation of the innate immune response, but further details of how this pathway is triggered are not provided. Therefore, due to the lack of novelty and mechanistic insight my feeling is that this manuscript is not really suited for EMBO Journal.

>Response: While it is known that ERV derepression can coincide with activation of the innate immune response (*Roulois et al., Cell 2015; Chiappinelli et al., Cell 2015*), it is not known which epigenetic factors are responsible for maintaining ERVs repressed in differentiated cells. The novelty in our manuscript is in the identification of KAP1 as one of these factors. As to the mechanism of how this ISG pathway is triggered, there is substantial evidence that in the case of 5-AZA treatment, it proceeds through MDA-5, MAVS and IRF7 (*Roulois et al., Cell 2015; Chiappinelli et al., Cell 2015*). We previously did not address the mechanism of how KAP1-depletion triggers an ISG response and now have done so in this revised version, thereby providing the enhanced novelty and mechanistic insight asked for:

In order to address the mechanism of how KAP1 depletion promotes innate immune activation, we considered that it could be a direct effect of KAP1 repression of ISGs through H3K9me3 recruitment. Or it could be an indirect effect in that upregulation of retrotransposons or other self-derived

nucleic acids in KAP1-depleted cells could be inducing either an RNA-sensing or DNA-sensing immune response. We addressed the first possibility by assessing the enrichment of KAP1 and H3K9me3 at ISGs compared to randomly-selected groups of genes (see Figure 5A), which showed no evidence for direct KAP1 regulation of ISGs. The corresponding text in the manuscript reads:

“We first asked if ISGs could be directly regulated by KAP1. For this we identified 437 ISGs (genes induced ten-fold upon IFN treatment), using the tool: <http://www.interferome.org> [38] and also

produced 3 independent sets of 437 randomly-selected genes. These genes (including 5kb each side) were intersected with H3K9me3 (18271 peaks[39]) or KAP1 (6148 peaks, from Figure S3D) ChIP-seq peaks, which showed no significant differences of KAP1 (present on 6% of ISGs) or silent chromatin (present on 14% of ISGs) per group (Figure 5A). This suggested KAP1 does not regulate ISGs directly.”

In order to address the second possibility that KAP1 depletion leads to an intrinsic innate immune response either by dsRNA or DNA sensing we employed a THP1-monocytic ISG reporter cell line (Figure 5DEF). We used CRISPR/Cas9 genome editing to knockout either MAVS or STING as the key adaptor proteins necessary for triggering an RNA-sensing vs. a DNA-sensing innate immune response, respectively. Results revealed that the modest ISG response that ensues from KAP1-depletion is dependent on MAVS not STING signalling and is therefore indicative of a dsRNAsensing response. This is an analogous response to what has been described for 5-AZA treatment (Roulois *et al.*, *Cell* 2015; Chiappinelli *et al.*, *Cell* 2015) and SETDB1 depletion (Cuellar *et al.*, *JCB* 2017).

Referee #2:

EMBOJ-2017-97480

In this manuscript, the authors described that the human endogenous retrovirus (ERV) HERVK14C is repressed by KAP1 in undifferentiated and differentiated cell lines. Furthermore, this pathway is functionally conserved in primary human peripheral blood mononuclear cells (PBMCs). They finally showed that cytosine methylation plays a role for silencing of KAP1-regulated ERVs and preventing production of immunostimulatory nucleic acids of the derepressed ERVs. Collectively, the authors proposed that the KAP1-KZNF pathway has evolved to play an important functional role in genome integrity and the control of viral mimicry in differentiated human cells. This is the first report to show that HERVK14C is derepressed by KAP1 knock down (KD) or knock out (KO) in differentiated human cells. Also, this is a novel finding that KD of KAP1 induces activation of interferon-stimulated genes (ISGs) in some human differentiated tumor cell lines. However, some of their evidences are not enough to support their entire ideas, such as how DNA methylation is crucial for KAP1-mediated HERVK14C silencing in PBMC or other tumor cell lines used in this study. Therefore, the reviewer requests to tackle following points to improve this manuscript for publication.

Major criticisms,

Q1. DNA methylation-KAP1 silencing issue

Bisulfite sequencing analysis should be done on the HERVK14C KAP1 binding or LTR regions in KAP1KO HeLa or 293T cells (Fig 1 E and F samples), and KAP1 KDed PBMCs and CD4+ T cells (Fig 5A and Fig S4C sample). DNA methylation and H3K9me3 analysis also should be done on the reporter constructs shown in Fig. 4B, C and 5B. Bioinformatics analysis showed that KAP1-targeted sites are enriched for H3K9me3 and SETDB1 (Fig 3A), but it is not demonstrated that how these epigenetic layers are functionally organized by KAP1 in differentiated cells. Once such data is available, the authors can address more mechanistic issue how KAP1 epigenetically represses ERV in differentiated cells.

>Response: We have now addressed the mechanism of how KAP1 regulates ERVs in differentiated cells. We first looked at DNA methylation as the reviewer suggested and focused on HeLa cells because in these cells we can document a clear effect of KAP1-depletion on expression of retrotransposons and ISGs (Figure 4). We measured DNA methylation at two KAP1-regulated retrotransposons, HERVK14C and SVAs. DNA methylation was highly enriched across SVA elements (Figure 4C) with a mean methylation of 73% in shControl cells. There was no significant change in DNA methylation in the KAP1-depleted cells (75% methylation), whereas demethylation was apparent in the 5-AZA treated cells, as expected. This result is consistent with previous data showing that KAP1 does not significantly impact on DNA methylation once established in embryonic cells either (Rowe *et al.*, *Nature* 2010). We did not measure DNA methylation in KAP1-depleted primary cells, since there was no effect in cell lines in which there was a more

pronounced phenotype of retrotransposon-regulation by KAP1. The HERVK14C LTR exhibited relatively low levels of DNA methylation compared to SVAs that was little impacted by KAP1-depletion or 5-AZA (Figure S6E). These data suggest that KAP1 does not regulate ERVs in differentiated cells through maintaining DNA methylation and instead may be mainly required to repress ERVs that have escaped DNA methylation control. We then looked at the potential impact of KAP1-depletion on H3K9me3 enrichment at retrotransposons: We found H3K9me3 to be enriched on HERVK14C, SVAs and ZNF genes in HeLa cells and 293T cells (Figure 5B). We then depleted KAP1 and SETDB1 in parallel in 293T cells (Figure 5C), showing that KAP1, like SETDB1 contributes to H3K9me3 maintenance in these cells.

Q2. ISG activation issue

KD of KAP1 induced ISG activation in HeLa cells but not in PBMCs. However, KAP1-targeted ERVs such as HERVKC14 and SVA D VNTR were derepressed in both KAP1 KDed and 5-aza treated PBMCs (Fig 5A and Fig6B bottom panels), derepression level was even less in 5-aza treated ones.

Thus, this data is inconsistent with the authors' simple story "5-aza treatment induces derepression of the KAP1-targeted ERVs by removing DNA methylation on those KAP1-targeted ERVs, then transcript of the derepressed ERVs induces ISG activation through ERV-derived nucleic acid sensing" shown in Fig 7. Thus, 5-aza has additional effect other than ERV derepression for the ISG activation. The authors should describe their results more carefully.

>**Response:** This is true and we have now changed our model (now Figure 6) to reflect the results that we have documented in this study and we have removed 5-AZA in the diagram because it is unclear if it acts through the same mechanism as KAP1-depletion. Note that although we have documented that KAP1 regulates both HERVs and SVAs, we do not know if these precise elements are responsible for the stimulation of an RNA-sensing immune response. We have therefore included "other endogenous RNAs?" in the scheme. Addressing which transposons do stimulate an innate immune response here is beyond the scope of this current work and forms the basis of a follow up project we are working on.

Minor points,

Q3. Fig. S2B,

The authors should discuss the mechanism of how these elements are repressed by KAP1 KO.

>**Response:** Now done, see page 6: "Of note, several retrotransposons were downregulated in knockout cells, which we attribute to indirect effects since we could not detect KAP1 binding to them (ENCODE) (Figure S2B)".

Q4. RNA expression analysis of shKAP1 treated cells, the authors should describe when samples are harvested after KD.

>**Response:** We have now ensured that all time points are stated for RNA expression analysis in shKAP1 cells: This was done in NTERA-2 cells (Figure 1C) where time points are labelled on the Figure, and in PBMCs (Figure 1F) where the time point was day 6 post transduction, now stated in the legend. Then Figure 4, Figure S1F and Figure S6 time-points were all day 6 as now stated in the legends.

Q5. Fig. S4A, right panel. Is this KAP1 KD efficiency data? If so, no significant reduction in this case based on B2M as a standard? The authors should clarify.

>**Response:** This data is now in Figure S1F. KAP1 depletion was much less efficient in PBMCs than in cell lines due to lower transduction efficiency of these cells, despite puro selection. Nevertheless, we can document a depletion of KAP1 in this experiment with GAPDH normalization, whereas it is not significant with B2M normalization and we do observe differences between normalization genes, which are not as apparent when expression differences are very big. Note that KD efficiency is here measured on the bulk population of cells so this result is not reflective of the KD efficiency within the population of cells most highly transduced. We have now stated this in the Figure S1F legend.

Q6. Fig. 5D This data should be shown in supplemental fig since just protein expression data does not mean for function of them in KAP1-mediated ERV silencing.

>**Response:** We have now moved this Western blot into Figure S5.

Q7. Fig. 6A left panel. Font side of the ERV labels is inconsistent with other panels.

>**Response:** Now corrected, see new Figure 4.

2nd Editorial Decision

11 June 2018

Thank you for the submission of your revised manuscript to our editorial offices. We have now received the reports from the two referees that were asked to re-evaluate your study (you will find enclosed below). As you will see, both referees have raised further concerns and/or suggestions, we ask you to address in a final revised manuscript.

We think it will be important to provide either more insight on how KAP1 controls ERV silencing via DNA-methylation, or to change your statements accordingly, as indicated by referee #1 in his major points 1-4. Please also address the other points of both referees in the revised manuscript, and in a point-by-point response.

Further, I have the following editorial requests:

- Your manuscript has currently 6 main figures, and 7 Appendix figures. As already mentioned in my previous decision letter, we now prefer to present important supplementary data in the Expanded View format (which will be displayed in the main HTML of the paper in a collapsible format). You can select up to 5 images from your Appendix as Expanded View, which we suggest to do. Please follow the nomenclature Figure EV1, Figure EV2 etc. in the manuscript text and the legends. The figure legend for these should be included in the main manuscript document file in a section called 'Expanded View Figure Legends' after the main 'Figure Legends'. Additional Supplementary material you can then put into an Appendix. Please provide the Appendix as one single pdf labelled 'Appendix'. The Appendix includes a table of content on the first page (with page numbers), then all figures, tables and their legends. Please follow the nomenclature Appendix Figure Sx (Appendix Table Sx) throughout the text, the Appendix TOC, their legends and the labels. For more details please refer to our guide to authors:
<http://embor.embopress.org/authorguide#manuscriptpreparation>

- Please move both tables as supplementary tables to the Appendix. I do not think it is necessary that these tables are shown in the online version of the article.

- It seems, there is currently no callout for Figure 5F in the text. Please add this.

- You provided 6 files as spreadsheets. It is unclear if these are datasets, source data, or should be part of the Appendix. Most likely spreadsheets 1-5 are datasets, and spreadsheet 6 (which is a text file) could go to the Appendix (as a table?). Please rename the files and upload these as dataset files (or include these in the Appendix if more appropriate). Finally, please their callouts in the main text accordingly.

- Regarding data quantification and statistics, can you please specify, where applicable, the number "n" for how many independent experiments (biological replicates) were performed, the bars and error bars (e.g. SEM, SD) and the test used to calculate p-values in the respective figure legends. Statistical testing only makes sense if n is >2. Please provide statistical testing where applicable. See also:
<http://embor.embopress.org/authorguide#statisticalanalysis>

- Please indicate in the legends for all bar diagrams what the numbers above some bars (e.g. 2.2x, 6.1x and 4.1x in Fig. 1C) mean.

- Please provide the Western Blot images with higher resolution, as unmodified as possible, and with similar contrast/brightness. E.g. both lower panels in Fig 5E are over-contrasted compared to the upper panel. Further, please provide the original source data for the Western blots. The source data will be published in a separate source data file online along with the accepted manuscript and

will be linked to the relevant figure. Please submit scans of entire blots, including size markers, label the scans with figure and panel number, and send one PDF file per figure.

- a Microsoft Word file (.doc) of the revised manuscript text
- editable TIFF or EPS-formatted EV figure files in high resolution
- the Appendix file

In addition I would need from you:

- a short, two-sentence summary of the manuscript
- two to three bullet points highlighting the key findings of your study
- a schematic summary figure (in jpeg or tiff format with the exact width of 550 pixels and a height of about 400 pixels) that can be used as a visual synopsis on our website.

REFEREE REPORTS

Referee #1:

The authors responded to most of the reviewer's comments and they are mostly fine. However, some of the key issues for this study and data are still not well addressed or described in the revised manuscript, especially KAP1-mediated vs DNA methylation-mediated ERV silencing. Therefore, the reviewer requests following additional points to respond for improving this work before publication.

Major points,

1) The authors explain the reason of modest impact of KAP1KD on the ERV derepression by some redundant mechanism with DNA methylation, such as "This may reflect redundant silencing mechanisms at these elements and in line with this, we found cytosine methylation to be enriched at SVA elements (Figure S1E)."p5

" Overall, these results suggest that KAP1 is an important gatekeeper of innate immune sensing of retrotransposons but in differentiated cells it exerts a modest effect likely because its partly redundant with cytosine methylation, for example at repressing SVAs."p11

If so, why additive effect was not seen for shKAP1+5-AZA treatment, such as shown in Fig. S6D. Data is inconsistent with the statement, thus the authors should provide supportive evidence or change the statement of this issue more appropriately.

2) Also, if the level of DNA methylation on the KAP-1-targeted ERVs are quite different in cell lines or cell types (for example, HERVK14C in HeLa vs CD4+ T cells) and this DNA methylation level is quite important for the response of KAP1KD, the authors should also clarify the DNA methylation status of them, at least two model ERV loci, HERVK14C and SVA in the examined cells other than HeLa.

3) The authors also state that

"This suggests that KAP1 may be mainly only required in differentiated cells to regulate ERVs that escape or exhibit dynamic DNA methylation, as proposed for SETDB1 [25]."p10

"In differentiated cells in contrast, cytosine methylation takes over as the main silencing mechanism and KAP1 may only be required where cytosine methylation is dynamic or absent, as proposed for SETDB1 [25]."p13

However, main point of reference 25 is opposite to the idea "cytosine methylation takes over as the main silencing mechanism in differentiated cells". Therefore, this citation is not appropriate.

Furthermore, as pointed in 1), the issue of KAP1-mediated and DNA methylation-mediated ERV silencing should be carefully discussed.

4) Based on the revised experiments, the authors commented the KAP1-DNA methylation issue in the response letter,

"These data suggest that KAP1 does not regulate ERVs in differentiated cells through maintaining DNA methylation and instead may be mainly required to repress ERVs that have escaped DNA methylation control. We then looked at the potential impact of KAP1-depletion on H3K9me3 enrichment at retrotransposons: We found H3K9me3 to be enriched on HERVK14C, SVAs and ZNF genes in HeLa cells and 293T cells (Figure 5B). We then depleted KAP1 and SETDB1 in parallel in 293T cells (Figure 5C), showing that KAP1, like SETDB1 contributes to H3K9me3 maintenance in these cells."

If so, the author also should show whether the same KAP1-targeted ERVs are derepressed by SETDB1KD in the examined cells, such as shown in Fig. 5C to complete the story.

Minor points,

5) related to the original reviewer's comment, Q5, still some figures, there is no description whether GAPDH or B2M was used for normalization in the figure legend. Thus, Fig. 4A, Fig. S1F and H, Fig. S6A-D. Should state it.

6) Fig. 1E right panel, need explanation of what is clone I.

7) Still statistics validation data are missing in some of data, such as Fig. 4B left panel, 4C shcontrol vs shKAP1 and DMSO vs 5-AZA, Fig. 5D left panel, 5F, Fig. S1C Bulk lanes, S1H, S2B, S6B, S6D-F (label F is missing).

8) Fig. 4B PBMCs panel, it is not clear whether this is shRNA or 5-AZA treatment experiment based on the legend, "(B) qRT-PCR expression of endogenous repeats (left) and ISGs (right) following 5-AZA treatment of HeLa cells and PBMCs (day 6 post transduction)". It should be clarified.

9) Fig. 5A inside overlay fig need explanation in the legend. What are these two circles? Also, Random 1 bar can't be recognized. It should be outlined.

10) Fig. 6. Need explanation of "red cross" on the arrow. Does this indicate inactivation of KAP1 function? Also, in legend, "KAP1 and SETDB1 binding are detected at ERVs and ZNF genes in differentiated cells and overlap with the silent chromatin mark H3K9me3 as well as cytosine methylation, which we detect at HERVK14C and SVAs.", but no own data of SETDB1 binding in this study. If describe like this, original work should be cited.

11) Fig. S6D, shKAP1+5-AZA treatment experiment, which day the samples were harvested. In the legend, sh-KD was done for 5-6 days and 5-AZA was done for 2 days. Furthermore, if harvesting time is different between two samples, is it relevant to compare them on the same panel for the level of ERV and ISG expression?

12) No explanation of how much amount of 5-AZA was used. Please state it. Furthermore, 5-azacytidine (5-AZA) should be stated at the first use.

13) Fig. S4A, lane "KSP1 common". They are 100% repeat elements (no genes)?

Referee #2:

Referees are asked to supply answers to the following questions, with brief accompanying comments where appropriate:

1. Does this manuscript report a single key finding? YES
Kap1 deletion in human cells results in ERV rerepression

2. Is the reported work of significance (YES), or does it describe a confirmatory finding or one that has already been documented using other methods or in other organisms etc (NO)? YES/NO
The work is of significance as it demonstrates that Kap1 is an important ERV silencing factor in human cells. However, the work is largely confirmatory as Kap1 deletion in human cells was done before.

3. Is it of general interest to the molecular biology community? YES/NO

ERV regulation is of general interest in the field of gene regulation, as ERVs may contribute to host gene regulation and, aberrant expression of ERVs correlates with diseases.

4. Is the single major finding robustly documented using independent lines of experimental evidence (YES), or is it really just a preliminary report requiring significant further data to become convincing, and thus more suited to a longer format article (NO)? YES

The finding that Kap1 regulates ERVs in human cells is robustly documented and in line with previous reports in mouse and human.

The major finding of this manuscript is that Kap1 silences specific ERV classes in human cells (e.g. HERVK14C). Furthermore, Kap1 knock-down leads to activation of an innate immune response, mainly dependent on the RNA detection pathway. The findings very well confirm the central function of Kap1 in silencing distinct ERV classes in human and mouse (e.g. PMIDs: 28052240, 29290627, 20075919).

The following points need to be considered:

1) The manuscript mainly describes the function of Kap1 in ERV silencing. This also involves detecting how Kap1 interplays with other silencing machineries, e.g. H3K9me3 and DNA methylation. In my opinion the manuscript would be easier to read if the logic of this interplay would be strengthened. In the current version of the ms, analysis of DNA methylation is coupled with activation of the ISG response. However, there is no established link between ERV derepression and ISG activation. Therefore I would suggest to document histone and DNA methylation changes (and no changes) in Kap1 ko cells separat from the ISG analysis.

2) For the analysis of ISG the authors use shKap1 in HeLa cells (Fig.4). Why did they not use the established ko clones shown in Fig. 1D)? shRNA knock-down may lead to side effects of treatment and off-target effects. In fact there appear to be severe side effects of shKap1 as HeLa cells die upon knock-down (Fig. S6F).

3) The growth defect of shKap1 in HeLa cells (Fig. S6F) cannot be attributed to selection processes upon ISG activation. This should be marked as pure speculation in the text.

1) It is not at all clear how Fig3E it connected to the rest of this figure. Either remove or explain better.

2nd Revision - authors' response

21 June 2018

Referee #1

The authors responded to most of the reviewer's comments and they are mostly fine. However, some of the key issues for this study and data are still not well addressed or described in the revised manuscript, especially KAP1-mediated vs DNA methylation-mediated ERV silencing. Therefore, the reviewer requests following additional points to respond for improving this work before publication.

Major points,

1) The authors explain the reason of modest impact of KAP1KD on the ERV derepression by some redundant mechanism with DNA methylation, such as "This may reflect redundant silencing mechanisms at these elements and in line with this, we found cytosine methylation to be enriched at SVA elements (Figure S1E)."p5

Author response: We don't know why SVAs are harder to resurrect and suggest their DNA

methylation may be important. We now change our above statement in the text quoted by the reviewer to make sure it's clear that this is not a proven fact as follows: "One possibility why SVAs may be harder to resurrect could be due to their enriched cytosine methylation (Fig EV1E)".

" Overall, these results suggest that KAP1 is an important gatekeeper of innate immune sensing of retrotransposons but in differentiated cells it exerts a modest effect likely because its partly redundant with cytosine methylation, for example at repressing SVAs."p11
Author response: We have now modified this sentence in the text to reflect the message of the paper and it now reads: 'Overall, these results suggest that KAP1 contributes to the regulation of retrotransposons and innate immune genes'.

If so, why additive effect was not seen for shKAP1+5-AZA treatment, such as shown in Fig. S6D. Data is inconsistent with the statement, thus the authors should provide supportive evidence or change the statement of this issue more appropriately.

Author response: These statements have now been changed and we have now removed Figure S6D since we cannot rule out a potential additive effect here with increased shKAP1 depletion or increased time of the two treatments.

2) Also, if the level of DNA methylation on the KAP-1-targeted ERVs are quite different in cell lines or cell types (for example, HERVK14C in HeLa vs CD4+ T cells) and this DNA methylation level is quite important for the response of KAP1KD, the authors should also clarify the DNA methylation status of them,

at least two model ERV loci, HERVK14C and SVA in the examined cells other than HeLa.
Author response: The message of this work is that KAP1 contributes to the regulation of retrotransposons and innate immune genes in human adult tissues. We further document that this mechanism involves maintenance of H3K9me3 at repeats and zinc finger genes, and an intrinsic RNA sensing response. In parallel we observe that 5-AZA has a more striking effect on repeats and ISGs than KAP1 depletion. However, we do not couple the role of KAP1 here to changes in DNA methylation and we have now changed any statements that may suggest that KAP1 exerts more effect at repeats that have only low levels of DNA methylation (see above). Therefore, it is not necessary to document DNA methylation of different repeats in different cell types. While this could be an interesting study, it is not the focus of this current work.

3) The authors also state that "This suggests that KAP1 may be mainly only required in differentiated cells to regulate ERVs that escape or exhibit dynamic DNA methylation, as proposed for SETDB1 [25]."p10
 "In differentiated cells in contrast, cytosine methylation takes over as the main silencing mechanism and KAP1 may only be required where cytosine methylation is dynamic or absent, as proposed for SETDB1 [25]."p13
 However, main point of reference 25 is opposite to the idea "cytosine methylation takes over as the main silencing mechanism in differentiated cells". Therefore, this citation is not appropriate. Furthermore, as pointed in 1), the issue of KAP1-mediated and DNA methylation-mediated ERV silencing should be carefully discussed.

Author response: We have now deleted these statements.

4) Based on the revised experiments, the authors commented the KAP1-DNA methylation issue in the response letter,

"These data suggest that KAP1 does not regulate ERVs in differentiated cells through maintaining DNA

methylation and instead may be mainly required to repress ERVs that have escaped DNA methylation

control. We then looked at the potential impact of KAP1-depletion on H3K9me3 enrichment at retrotransposons: We found H3K9me3 to be enriched on HERVK14C, SVAs and ZNF genes in HeLa cells

and 293T cells (Figure 5B). We then depleted KAP1 and SETDB1 in parallel in 293T cells (Figure 5C),

showing that KAP1, like SETDB1 contributes to H3K9me3 maintenance in these cells."

If so, the author also should show whether the same KAP1-targeted ERVs are derepressed by SETDB1KD in

the examined cells, such as shown in Fig. 5C to complete the story.

Author response: Yes, we verified that SETDB1 depletion also affects ERV repression (3 fold upregulation of HERVK14C in SETDB1-kd cells compared to 5 fold upregulation of HERVK14C in KAP1-kd here). This plot is now added to Figure 5 (Fig 5B).

Minor points,

5) related to the original reviewer's comment, Q5, still some figures, there is no description whether GAPDH

or B2M was used for normalization in the figure legend. Thus, Fig. 4A, Fig. S1F and H, Fig. S6A-D. Should

state it.

Author response: now stated in legend

6) Fig. 1E right panel, need explanation of what is clone I.

Author response: now stated in legend

7) Still statistics validation data are missing in some of data, such as **Fig. 4B left panel**, 4C shcontrol vs

shKAP1 and DMSO vs 5-AZA, Fig. 5D left panel, 5F, Fig. S1C Bulk lanes, S1H, S2B, S6B, S6D-F (label F

is missing).

Author response: now stated in legend

8) Fig. 4B PBMCs panel, it is not clear whether this is shRNA or 5-AZA treatment experiment based on the

legend, "(B) qRT-PCR expression of endogenous repeats (left) and ISGs (right) following 5-AZA treatment

of HeLa cells and PBMCs (day 6 post transduction)". It should be clarified.

Author response: now clarified on figure

9) Fig. 5A inside overlay fig need explanation in the legend. What are these two circles? Also, Random 1

bar can't be recognized. It should be outlined.

Author response: now clarified in legend

10) Fig. 6. Need explanation of "red cross" on the arrow. Does this indicate inactivation of KAP1 function?

Also, in legend, "KAP1 and SETDB1 binding are detected at ERVs and ZNF genes in differentiated cells

and overlap with the silent chromatin mark H3K9me3 as well as cytosine methylation, which we detect at

HERVK14C and SVAs.", but no own data of SETDB1 binding in this study. If describe like this, original

work should be cited.

Author response: The red cross is now replaced with "KAP1 inactivation". In this work, we analyse

SETDB1 binding at KAP1 bound sites using ENCODE data (Fig 2F, paper now referenced in Fig 6

legend) and show SETDB1 regulates H3K9me3 at ERVs which is necessary for their repression (Fig 5C).

11) Fig. S6D, shKAP1+5-AZA treatment experiment, which day the samples were harvested. In the legend, sh-KD was done for 5-6 days and 5-AZA was done for 2 days. Furthermore, if harvesting time is different between two samples, is it relevant to compare them on the same panel for the level of ERV and ISG expression?

Author response: This panel is now removed. However, we did need to do the experiment this way because 5-AZA is toxic after more than 2 days and KAP1 depletion takes 5-6 days to induce a phenotype so we usually add 5-AZA for the two consecutive days before the KAP1 time-point of day 6 post transduction.

12) No explanation of how much amount of 5-AZA was used. Please state it. Furthermore, 5-azacytidine (5-AZA) should be stated at the first use.

Author response: Now stated in legend and methods

13) Fig. S4A, lane "KSP1 common". They are 100% repeat elements (no genes)?

Author response: Here we took all LINE1 and ERV coordinates and intersected them with KAP1 peaks (614 common peaks). Some peaks overlap multiple repeats and some may overlap no repeats. Genes were not analysed in this figure (but are in Fig 2E instead). The precise identity of each KAP1 peak in terms of its overlap with genes and repeats is given in Dataset 3. We now better explain this in the legend (now Fig EV3A).

----- Referee #2

Referees are asked to supply answers to the following questions, with brief accompanying comments where appropriate:

1. Does this manuscript report a single key finding? YES
Kap1 deletion in human cells results in ERV rerepression
2. Is the reported work of significance (YES), or does it describe a confirmatory finding or one that has already been documented using other methods or in other organisms etc (NO)? YES/NO
The work is of significance as it demonstrates that Kap1 is an important ERV silencing factor in human cells.
However, the work is largely confirmatory as Kap1 deletion in human cells was done before.
3. Is it of general interest to the molecular biology community? YES/NO
ERV regulation is of general interest in the field of gene regulation, as ERVs may contribute to host gene regulation and, aberrant expression of ERVs correlates with diseases.
4. Is the single major finding robustly documented using independent lines of experimental evidence (YES), or is it really just a preliminary report requiring significant further data to become convincing, and thus more suited to a longerformat article (NO)? YES
The finding that Kap1 regulates ERVs in human cells is robustly documented and in line with previous reports in mouse and human.

The major finding of this manuscript is that Kap1 silences specific ERV classes in human cells (e.g. HERVK14C). Furthermore, Kap1 knock-down leads to activation of an innate immune response, mainly

dependent on the RNA detection pathway. The findings very well confirm the central function of Kap1 in silencing distinct ERV classes in human and mouse (e.g. PMIDs: 28052240,29290627, 20075919).
Author response: The above PMIDs link to manuscripts documenting the role of KAP1 in mouse ESCs (20075919) and human ESCs (29290627) (we now reference this work) and in human development in neural progenitor cells (28052240). The aim of the current work was to assess the role of KAP1 in human adult tissues. This is relevant to KAP1 safeguarding of the host innate immune response in people. We have now changed the title of our manuscript to reflect the difference of this present work to previous studies. 'KAP1 regulates ERVs in adult human cells and contributes to innate immune control'. The running title too has been changed to "KAP1 relevance in adult human cells".

The following points need to be considered:

1) The manuscript mainly describes the function of Kap1 in ERV silencing. This also involves detecting how Kap1 interplays with other silencing machineries, e.g. H3K9me3 and DNA methylation. In my opinion the manuscript would be easier to read if the logic of this interplay would be strengthened. In the current version of the ms, analysis of DNA methylation is coupled with activation of the ISG response. However, there is no established link between ERV derepression and ISG activation. Therefore I would suggest to document histone and DNA methylation changes (and no changes) in Kap1 ko cells separat from the ISG analysis.

Author response: We now make clear that there is no known link between ERV upregulation and the ISG response (see page 14 'Note, however, that to date there is no direct link known between ERV upregulation and an ISG response.'). We have considered all ways of rearranging the figures and have finally moved the ISG-related panel from Fig 5A to appear directly before the ISG-related data in Fig 5D.

2) For the analysis of ISG the authors use shKap1 in HeLa cells (Fig.4). Why did they not use the established ko clones shown in Fig. 1D)? shRNA knock-down may lead to side effects of treatment and off-target effects. In fact there appear to be severe side effects of shKap1 as HeLa cells die upon knock-down (Fig. S6F).

Author response: We did use ko clones and these clones had ISGs strongly downregulated. These cells took several months to single-cell clone and an ISG response is not usually indefinite particularly if it leads to apoptosis. We therefore looked at kd cells as well and found ISGs upregulated, a result we have repeated more than 5 times. This result was validated with two independent hairpins (Fig 4A) and a third independent hairpin in the first authors thesis making off-target effects unlikely.

3) The growth defect of shKap1 in HeLa cells (Fig. S6F) cannot be attributed to selection processes upon ISG activation. This should be marked as pure speculation in the text.

Author response: We now clarify that negative phenotypes would be outselected: Page 10: 'Of note, negative phenotypes would have been outselected during single-cell cloning of knockout clones and we found that initial KAP1-depletion caused a growth defect (Fig S2E)'.

1) It is not at all clear how Fig3E is connected to the rest of this figure. Either remove or explain better.

Author response: We now link Fig 3E better with an extra sentence following on from Fig 3D: Page 9:

'This suggested that a subset of KZNFs must be expressed in multiple differentiated cell types to recruit KAP1 to ERVs.'

Accepted

9 July 2018

I am very pleased to accept your manuscript for publication in the next available issue of EMBO reports. Thank you for your contribution to our journal.

At the end of this email I include important information about how to proceed. Please ensure that you take the time to read the information and complete and return the necessary forms to allow us to publish your manuscript as quickly as possible.

As part of the EMBO publication's Transparent Editorial Process, EMBO reports publishes online a Review Process File to accompany accepted manuscripts. As you are aware, this File will be published in conjunction with your paper and will include the referee reports, your point-by-point response and all pertinent correspondence relating to the manuscript.

If you do NOT want this File to be published, please inform the editorial office within 2 days, if you have not done so already, otherwise the File will be published by default [contact: emboreports@embo.org]. If you do opt out, the Review Process File link will point to the following statement: "No Review Process File is available with this article, as the authors have chosen not to make the review process public in this case."

Should you be planning a Press Release on your article, please get in contact with emboreports@wiley.com as early as possible, in order to coordinate publication and release dates.

Thank you again for your contribution to EMBO reports and congratulations on a successful publication. Please consider us again in the future for your most exciting work.

REFeree REPORT

Referee #1:

The manuscript is suitable for publication in EMBO reports without revision. Have no further comments.

Corresponding Author Name: Helen M Rowe

Manuscript Number: EMBOR-2017-45000V2